# Neurogaming in Virtual Reality: A Review of Video Game Genres and Cognitive Impact

Jesus GomezRomero-Borquez [1,†] , Carolina Del-Valle-Soto [1,*,†] , J. Alberto Del-Puerto-Flores [1] , Ramon A. Briseño [2] and José Varela-Aldás [3]

[1] Facultad de Ingeniería, Universidad Panamericana, Álvaro del Portillo 49, Zapopan 45010, Mexico; jagomez@up.edu.mx (J.G.-B.); jpuerto@up.edu.mx (J.A.D.-P.-F.)
[2] Centro Universitario de Ciencias Económico Administrativas, Universidad de Guadalajara, Zapopan 45180, Mexico; alejandro.bmartinez@alumnos.udg.mx
[3] Centro de Investigaciones de Ciencias Humanas y de la Educación—CICHE, Universidad Indoamérica, Ambato 180103, Ecuador; josevarela@uti.edu.ec
* Correspondence: cvalle@up.edu.mx; Tel.: +52-3313682200
† These authors contributed equally to this work.

**Abstract:** This work marks a significant advancement in the field of cognitive science and gaming technology. It offers an in-depth analysis of the effects of various video game genres on brainwave patterns and concentration levels in virtual reality (VR) settings. The study is groundbreaking in its approach, employing electroencephalograms (EEGs) to explore the neural correlates of gaming, thus bridging the gap between technology, psychology, and neuroscience. This review enriches the dialogue on the potential of video games as a therapeutic tool in mental health. The study's findings illuminate the capacity of different game genres to elicit varied brainwave responses, paving the way for tailored video game therapies. This review contributes meaningfully to the state of the art by offering empirical insights into the interaction between gaming environments and brain activity, highlighting the potential applications in therapeutic settings, cognitive training, and educational tools. The findings are especially relevant for developing VR gaming content and therapeutic games, enhancing the understanding of cognitive processes, and aiding in mental healthcare strategies.

**Keywords:** video games; cognitive and affective states; virtual reality; neurogaming

## 1. Introduction

Video games are interactive digital entertainment experiences that have evolved into a diverse medium with a wide range of applications in contemporary society. They serve not only as a source of entertainment and leisure but also as powerful tools for education, training, and simulation. Video games are employed in various educational settings to enhance learning outcomes, promote problem-solving skills, and facilitate immersive experiences. They are also utilized in professional training, ranging from healthcare simulations to military exercises, to improve skills and decision-making in high-stress environments [1]. Furthermore, video games have become a platform for social interaction, connecting individuals across the globe in massive online communities. In the realm of healthcare, they are harnessed for therapeutic purposes, aiding in physical rehabilitation and cognitive treatments.

Video games have undergone a remarkable transformation from their origins as simple pixelated diversions to become a cultural phenomenon that extends far beyond the confines of entertainment. While video games were once seen as mere pastimes for adolescents, they have since evolved into a multi-billion-dollar industry with profound implications for fields as diverse as healthcare and education [2]. In 2018 alone, the total revenue generated from the sales and distribution of video games on a global scale underwent a significant increase of 47%, showcasing a substantial growth trajectory in this particular

sector compared with other types of industries, as shown in Figure 1. Such a substantial rise in revenue clearly indicates the profound economic implications and influence that the video game industry exerts on a global scale, underscoring its pivotal role in shaping and contributing to the overall economic landscape of various countries and regions across the world. This transformation is largely attributed to the rapid advancements in technology that have granted video games the power to engage players on previously unimaginable intellectual, emotional, and physical levels.

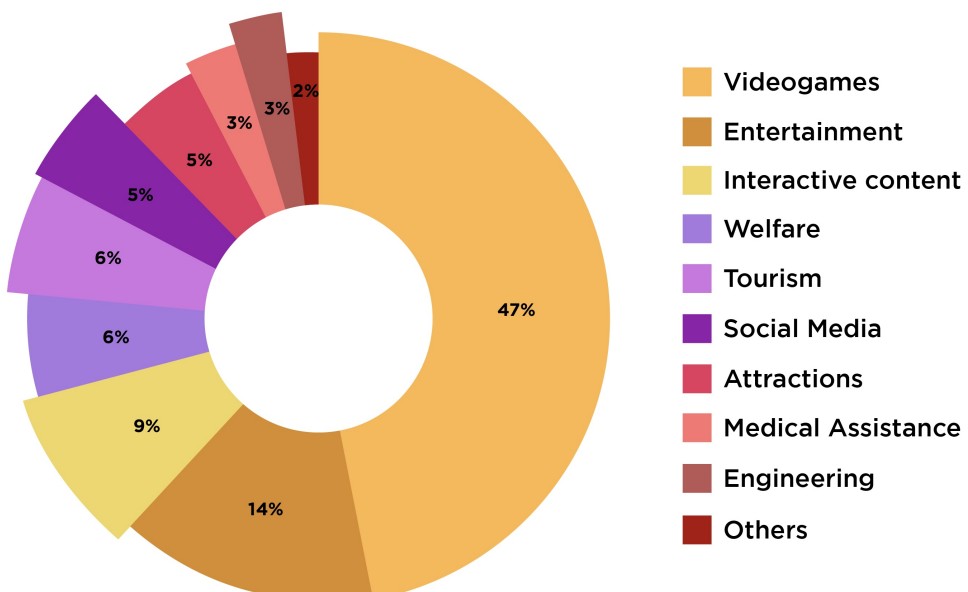

**Figure 1.** 2023 Revenues totaling $4.800 M.

The healthcare sector, in particular, has recognized the potential of harnessing video games as a tool for improving both physical and mental well-being. This recognition comes at a time when global health challenges, including the rising rates of chronic diseases and mental health disorders, are demanding innovative approaches to prevention, treatment [3], and rehabilitation. Video games, with their interactive nature and ability to captivate and motivate users, have emerged as a promising avenue for addressing these challenges [3].

The utilization of EEGs proves to be an influential instrument in acquiring significant data pertaining to video games. By enhancing our comprehension of the brain's operations while engaging in gameplay, we can procure valuable perceptions regarding the efficacy, enjoyment, and advantageous nature of various games for players. Brainwaves in EEGs provide valuable information about the state and activity of the brain. EEG signals can be used to analyze cognitive function, detect medical disorders such as epilepsy, and understand brain activity during surgical procedures [4]. EEG signals are divided into different frequency bands, including alpha, beta, gamma, theta, and delta, which can be used to estimate the current situation of the brain and determine whether the subject is in a state of pleasure or stress [5]. The analysis of EEG activity involves the summation of the synchronous activity of cortical neurons, providing insights into the temporal dynamics of the brain [6]. Various signal processing techniques, such as wavelet transform and statistical approaches, can be applied to analyze EEG data and gain a further understanding of the electrical activity in the brain [7,8].

EEG data, when analyzed in the frequency domain, reveal vital information about cognitive and affective states. Delta, theta, alpha, beta, and gamma waves, each representing different aspects of neural activity, play a crucial role in understanding how gamers engage with virtual reality [9]. Increased coherence in beta bandwidth, for instance, signifies high arousal when exposed to emotionally charged stimuli. The presence of theta waves indicates relaxed or meditative states and their modulation during transitions in

affective states. The dynamic relationships between pairs of EEG oscillations, such as phase synchronization and coherence, offer insights into affective arousal [10]. Alpha waves, on the other hand, vary with cognitive exertion and relate to valence and affective states, providing a window into the emotional responses of gamers. Beta, characterized by a low amplitude, is linked to cognitive processes like thinking and concentration, while gamma rhythms reflect the binding of neural networks for specific cognitive functions. The study mentioned in the provided text illustrates the complexity of EEG data analysis and the potential to gain a nuanced understanding of user experiences in VR gaming, setting the stage for further advancements in this exciting field.

Figure 2 provides a comprehensive and analytical overview of the intricate interplay between various video game genres and their corresponding impacts on brainwave patterns and cognitive states. It categorizes video games into genres such as fighting, shoot 'em up, platform, simulators, sports, strategy, adventure, role-playing, and educational, linking them to five distinct brainwave patterns: delta, theta, alpha, beta, and gamma waves. These brainwaves are indicative of varying cognitive and emotional states like relaxation, meditation, concentration, cognitive processing, and high cognition, which are influenced by video game interaction. Furthermore, the diagram explores the realm of virtual reality (VR) in gaming, highlighting its role in creating immersive experiences that demand physical engagement and active decision-making. Additionally, it touches upon the therapeutic uses of video games in stress reduction, social interaction, cognitive enhancement, mental health treatment, and rehabilitation, thereby illustrating the multifaceted impact of video games on both the brain and overall well-being.

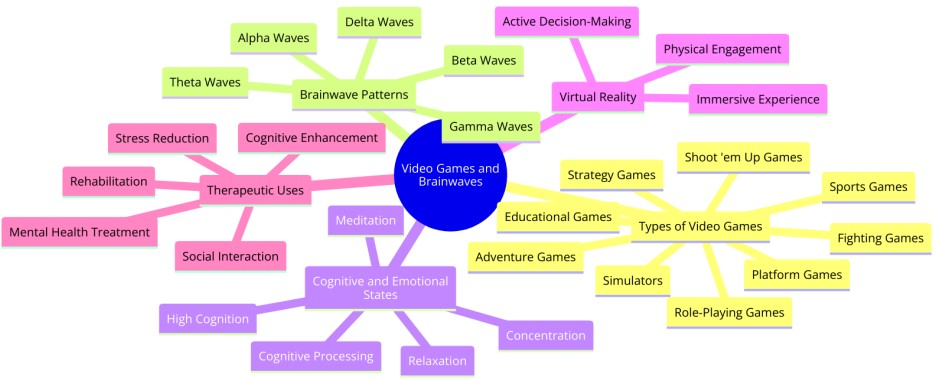

**Figure 2.** Mapping the impact: video game genres, brainwave patterns, and cognitive effects.

One of the most remarkable facets of video games' impact on healthcare is their ability to seamlessly merge entertainment with therapeutic benefits. Gamification, the process of applying game design elements to non-gaming contexts, has opened up new avenues for promoting healthy behaviors and managing chronic conditions [11]. From encouraging physical activity through exergames to providing cognitive therapy for individuals with neurological disorders, video games have become versatile tools in the healthcare toolkit.

Moreover, the concept of "serious games" has gained prominence, referring to video games explicitly designed for educational, training, or therapeutic purposes [12]. These games are not merely about entertainment; they are about achieving specific outcomes. Researchers and developers are increasingly collaborating to create serious games that target everything from teaching medical procedures to improving mental resilience. The intersection of gaming and healthcare is a dynamic space where innovation knows no bounds.

Figure 3 displays a categorization of video games that presents a fundamental challenge due to the issues of completeness and precise definitions in both content and form. This challenge is amplified when dealing with video games due to their vast versatility, thematic variety, and rich blend of proposals and genres. Classifying some games into exclusive categories becomes difficult, as many are sophisticated mixtures of adventure, strategy, combat, and competition. The categories discussed include fighting games, beat 'em up games, shoot

'em up games, platform games, simulations, sports games, strategy games, adventure games, role-playing games, war games, system simulators, board games, and educational games. Each category serves distinct purposes and involves different gameplay elements, making video game classification a complex endeavor. It is important to note that the boundaries between these categories can be blurred, and video games may possess educational potential and impact various cognitive and motor skills beyond their designated genres.

1.  Fighting games: Featuring hand-to-hand combat between player-controlled characters, often against computer opponents, with progressively changing environments, these games are known for explicit violence and bloodshed with realistic graphics and intense action.
2.  Beat 'em up (brawler) or combat games: Often confused with fighting games due to extreme violence, these games allow players to take on predetermined characters in urban settings, eliminating adversaries rapidly, usually with no deeper narrative or justification for violence.
3.  Shoot 'em up (shooter) games: Known for their intense violence, these games require players to shoot relentlessly at everything on the screen, with targets ranging from humanoid foes to alien invaders and menacing robots, promoting a focus on killing and destruction.
4.  Platform games: Typically centered on a character's quest through challenging landscapes, often involving rescuing a princess, collecting power-ups, and overcoming obstacles, these games may also include maze-like scenarios and hidden passages.
5.  Simulators: Realistic simulations of various activities, such as driving, flying, and more, they offer immersive experiences. They are often found in arcades, and they often focus on recreating real-world scenarios.
6.  Sports games: Based on real sports like football, golf, or basketball, they tend to be less violent and are suitable for multiplayer experiences. They have versatile themes and less controversial content.
7.  Strategy games: In this genre, players take on a specific identity with defined objectives, relying on tactics to achieve a successful outcome. It includes subcategories like adventure games, role-playing games (RPGs), war games, and system simulators (sims).
8.  Board games: Adaptations of classic board games like chess, checkers, and Scrabble, primarily found on personal computers, offer a less violent, intellectual gaming experience.
9.  Edutainment: These games that combine play with educational content, fostering learning through interactive activities, are categorized by structure, educational objectives, cognitive activities, and didactic strategies.

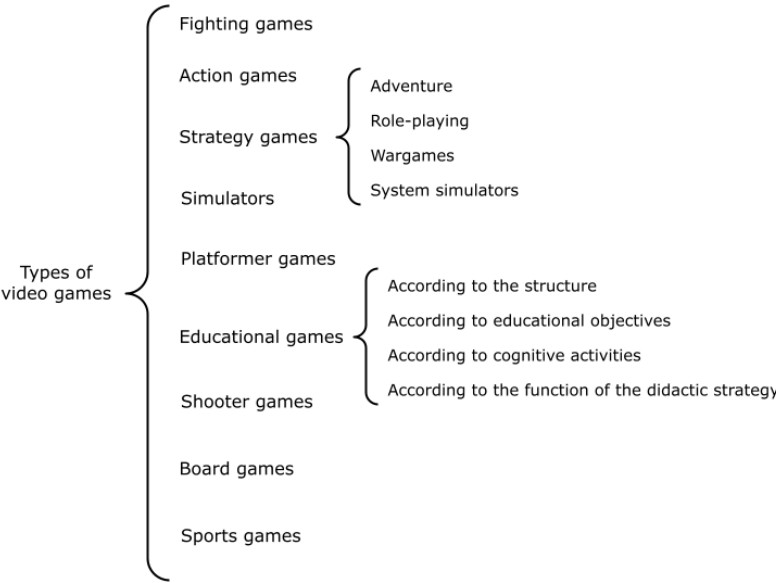

**Figure 3.** Classification of video games.

*Motivation*

Our review focuses on clarifying which types of video games have a more significant impact on inducing feelings of relaxation, profound emotions, heightened attention, increased alertness, and enhanced concentration levels to support cognitive functions. The study presents a compelling case for the intersection of video game genres and their profound impact on brainwave patterns, concentration levels, and mental health. This work is pivotal in advancing the state of the art, meticulously examining the cognitive and emotional responses elicited by various video game genres in virtual reality environments. Employing electroencephalogram technology, this review navigates uncharted territories in understanding the neural correlates of gaming. It contributes not only to cognitive science and neurogaming but also illuminates the therapeutic potential of video games in mental health, providing empirical insights into their application to mental disorders. This review is especially noteworthy, as it spans a comprehensive period from 2000 to 2023, offering a current and quantitatively analyzed perspective on neurogaming. Its findings suggest a transformative role for video games in healthcare and education, signifying a novel amalgamation of technology and medicine for cognitive enhancement and rehabilitation.

## 2. Related Work

Video game-related work with the use of brain waves has been explored in several studies. Researchers have integrated the use of brain waves as input controls in video games, allowing users to control the game using their brain activity [13]. Specific mechanics in video games have been designed to promote cognitive abilities such as short-term memory, attention, and concentration, and the activation of EEG waves during these mechanics has been analyzed [14]. In video games, eye movement techniques have also been used to stimulate different eye movement modes and explore the relationship between eye movements and brain wave activities [15]. Additionally, game apparatus and methods have been developed to detect and analyze brain wave signals, allowing for the control of games based on the user's attention and meditation levels [16]. Virtual reality game design technology involving brain waves has also been developed to control game content based on the user's brain wave information and detect the user's mental state [17].

Figure 4 shows the correlation between the level of acceptance and the length of time of the most well-received video games. Metascore [18] quantifies acceptance evaluation, representing an average rating of video game analyses assembled by Metacritic. The duration is quantified using statistics amassed by How Long to Beat [19], a platform offering information concerning video game length.

Metascore denotes a weighted mean of evaluations from critics for video games, which is compiled by Metacritic, a website dedicated to reviewing video games. The computation of the score involves an algorithm that ascribes a greater value to appraisals from prominent sources.

Metascore should not be considered a reliable gauge of a video game's popularity. It serves as a means of assessing the overall viewpoint of critics regarding a game, and it facilitates the comparison of different games.

Moreover, Metascore can be employed to anticipate the commercial success of a video game. Generally, games with higher Metascore ratings exhibit greater success than those with lower ratings.

How Long to Beat is an online platform that compiles and analyzes data pertaining to the length of video games. The information is gathered from diverse sources, encompassing video game evaluations, online forums, and individual player statistics. How Long to Beat is a valuable data repository regarding the duration of video games.

In addition, this website offers a means to conduct comprehensive investigations pertaining to video game length. The data obtained through this platform enable the identification of patterns concerning the duration of video games and facilitate comparisons across various genres and platforms.

The figure exhibits the influence of video games based on their classifications and Metascore assessments. The diagram is partitioned into four quadrants, each representing a distinct video game genre: emotional, physical, puzzle, and virtual reality. The most prevalent video games were selected within each genre.

Emotional video games encompass those that concentrate on storytelling and the exploration of emotions. The video games Limbo [20] (Playdead in Copenhagen, Denmark), Gris [21] (Nomada Studio based in Barcelona, Spain), Celeste [22] (Maddy Makes Games, now known as Extremely OK Games, Ltd., located in Vancouver, BC, Canada), Omori [23] (OMOCAT, LLC, situated in Los Angeles, CA, USA), and Disco Elysium [24] (ZA/UM in London, UK) are in this classification.

Physical video games require a considerable amount of dexterity and motor coordination. This genre includes the video games Nintendo Switch Sports [25], Wii Sports Resort [26], (Nintendo Switch Sports and Wii Sports Resort were both developed by Nintendo in Kyoto, Japan) Sports Champions [27] (San Diego Studio in San Diego, CA, USA), Wii Sports [28] (Nintendo in Kyoto, Japan), and ESPN International Track & Field [29] (Konami in Tokyo, Japan).

Puzzle video games necessitate the utilization of logical reasoning and problem-solving skills. This category includes the video games Flip Wars [30] (Over Fence Co., Ltd., location unknown), Neverout [31] (Setapp Sp. z o.o., location unknown), One Line Coloring [32] (MythicOwl in Warsaw, Poland), Human Resource Machine [33] (Tomorrow Corporation, location unknown), and TENS! [34] (Kwalee Ltd. in Leamington Spa, UK).

Virtual reality video games employ virtual reality technology to create an immersive gaming experience. Within this genre, one can find the video games Beat Saber [35] (Beat Games in Prague, Czech Republic), Five Nights at Freddy's: Help Wanted [36] (Steel Wool Studios in San Francisco, CA, USA), Among Us VR [37] (Innersloth in Redmond, WA, USA), PowerWash Simulator VR [38] (FuturLab in Brighton, UK), and BONELAB [39] (Stress Level Zero in Los Angeles, CA, USA).

The figure displays that emotional and virtual reality video games have the most profound impact on players. These video games are highly regarded for their narratives, aesthetics, challenges, novelty, and ability to provide a distinctive gaming encounter.

Table 1 provides a comprehensive overview of various research papers that investigate the impact of different types of video games on learning and development variables [40]. These studies explore how video games, ranging from educational and simulation games to action and role-playing games, can influence cognitive and socioemotional development, academic achievement, problem-solving skills, and other learning outcomes. The table also highlights the diverse usage of these games, from classroom learning and professional training to rehabilitation therapy and after-school programs. Independent variables such as game exposure, virtual reality gaming, and gaming device types are examined to understand their influence on the learning and development processes. This review underscores the potential of video games not only as tools for educational purposes but also as valuable resources for enhancing cognitive abilities, social skills, and even physical fitness, depending on the game type and context of use. It demonstrates the multifaceted role that video games can play in addressing emotional disorders, relaxation, and concentration in individuals, making them a versatile tool for promoting well-being and personal growth.

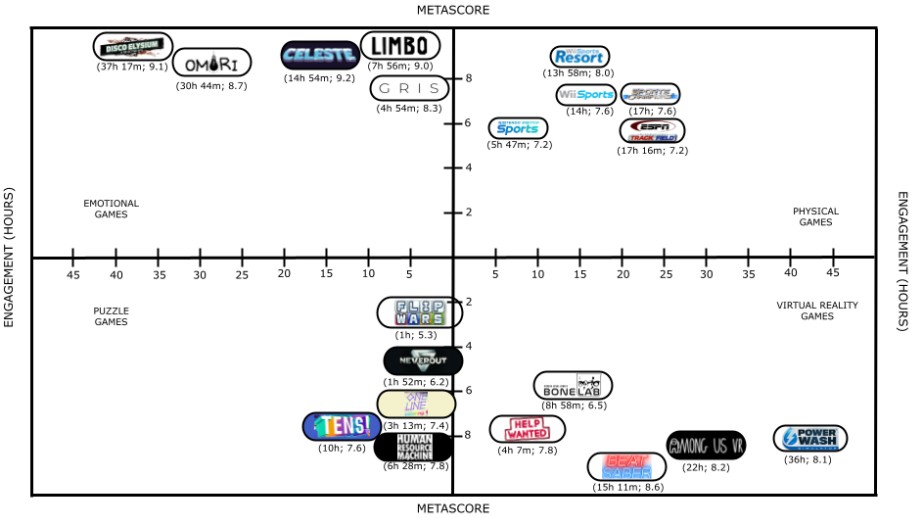

**Figure 4.** The impact of video games and their performance in the industry.

**Table 1.** Research papers on video games and learning.

| Reference | Game Type | Learning/Development Variables | Game Usage | Independent Variables |
|---|---|---|---|---|
| [41] | Educational | Cognitive development, academic performance | Classroom learning | Computer-based learning |
| [42] | Action | Socioemotional development | Entertainment | Video game exposure |
| [43] | Simulation | Academic achievement, problem-solving skills | Educational tool | Virtual reality gaming |
| [44] | Puzzle | Cognitive skills, memory retention | Brain training | Television viewing habits |
| [45] | Role-playing | Social skills, teamwork | Online gaming communities | Age and gender |
| [46] | Adventure | Language acquisition, creativity | After-school programs | Internet usage |
| [47] | Strategy | Critical thinking, decision-making | Professional training | Gaming device type |
| [48] | Sports | Physical fitness, hand–eye coordination | Physical education | Parental involvement |
| [49] | Racing | Reaction time, spatial awareness | Rehabilitation therapy | Gaming experience |
| [50] | Puzzle | Problem-solving skills, attention span | Mobile learning | Game genre preferences |
| [51] | Educational | Literacy skills, motivation | Home learning | Parental education level |
| [52] | Simulation | Environmental awareness, teamwork | Science education | Game duration |
| [53] | Adventure | Cultural awareness, empathy | Diversity education | Socioeconomic status |
| [54] | Strategy | Leadership skills, conflict resolution | Military training | Previous gaming experience |
| [55] | Role-playing | Ethical decision-making, empathy | Counseling sessions | Game narrative complexity |

Figure 5 illustrates the main research categories for experimentation in video games along with their measurement attributes. The first category focuses on the relationship between the use of educational apps and learning. The literature suggests that the use of educational apps significantly impacts the learning of various academic skills, including mathematics, geometry, English, geography, art, and literacy, as well as learning styles and motivation [56]. The second category explores the use of socioemotional and behavioral development. It includes variables like psychological well-being, anxiety, emotional comprehension, interpersonal relationships, prosocial behavior, social norms, and self-regulation [57]. The third category delves into the relationship between the use of apps and neuropsychological aspects such as graphomotor skills, intelligence, executive function, attention, working memory, language, and visuospatial skills [58]. The last category focuses on the impact of screen time on health aspects. Pre-experimental studies demonstrate the benefits of using digital tools to manage anxiety, resulting in improved attention, inhibitory control, and academic performance. Educational apps have also been found to enhance math skills and the number of problems solved, especially when presented with visual and auditory stimuli [59]. The effectiveness of language learning apps depends on cognitive load and task attitudes. Experimental studies highlight the impact of app-based neurofeedback on emotional self-regulation and mindfulness and the significant effects of apps on cognitive skills and specific knowledge areas, such as math and language.

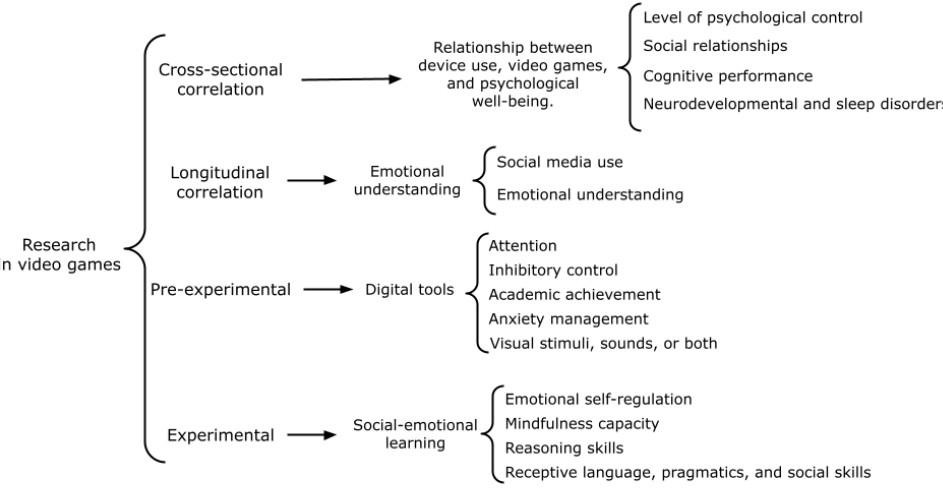

**Figure 5.** Research criteria in video game research.

Figure 6 exhibits the percentage of video games that promote cognitive aspects. Video games that promote fact memorization require players to remember information, such as historical facts or scientific data. Video games that promote logical and reasoning skills require players to solve problems using logic and reasoning. Video games that encourage problem-solving and strategic planning require players to plan and execute strategies in order to solve problems. Video games that promote scientific knowledge are those that teach players about scientific concepts. Video games that promote an increased focus require players to concentrate on a task for an extended period. Video games encouraging attention to detail require players to focus on small details. Video games that promote reading comprehension and vocabulary require players to read and understand text. Video games that promote spatial perception and recognition require players to have a good understanding of space and the ability to recognize objects. Video games that promote inductive discovery require players to make inferences from the information they are given. Video games that promote an increased ability to use symbols require players to use symbols to represent concepts. The figure provided visually depicts the fact that a significant proportion of video games are designed to enhance cognitive functions and skills. This visual representation implies that video games possess the potential to serve as

an effective instrument for facilitating the process of acquiring knowledge and fostering cognitive growth and advancement.

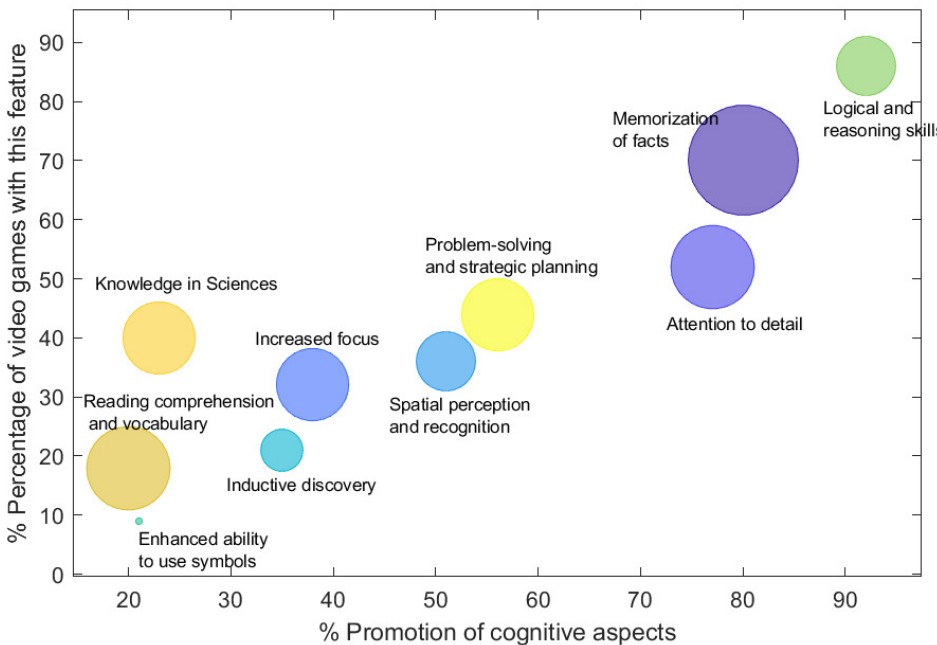

**Figure 6.** Cognitive aspects of video games.

## 3. Materials and Methods

Video games are being increasingly employed for the treatment of a diverse range of physical disorders. This is because video games have the capability to offer a secure, engaging, and stimulating method to enhance physical activity, enhance motor skills and coordination, and manage symptoms of diseases.

The flow diagram depicted in Figure 7 represents a comprehensive methodology for studying the impact of video games on brainwave patterns and cognitive states. It begins with the classification of video games, in which different types and categories are identified. This leads to an analysis of brainwave patterns using EEGs, focusing on delta, theta, alpha, beta, and gamma waves. The next phase assesses the impact of these patterns on cognitive and emotional states, considering factors like cognitive well-being and psychological assessment. The investigation then delves into the impact of VR games on brainwave activity, analyzing effects on cognitive functions and concentration levels. This involves examining concentration levels and overall brain activity. The methodology further explores the use of video games in a rehabilitation context, looking into therapeutic applications and health benefits. Finally, it culminates in a discussion on the implications of VR and neurogaming, focusing on VR technology trends and neurogaming developments, before concluding the study.

The class diagram, depicted in Figure 8, presents a comprehensive mapping of critical concepts derived from the paper, showcasing the intricate interplay among various elements. At its core, the diagram features the "VideoGame" class, elaborating its fundamental attributes: Title, Genre, Platform, Purpose, Modes, and AdaptedFeatures. This class is directly linked to "BrainwavePattern", highlighting the influential role video games play in modulating brainwave activities, such as delta, theta, alpha, beta, and gamma waves. The "CognitiveEffect" and "EmotionalState" classes further extend the impact of video games, capturing their effect on mental processes and emotional well-being, as underscored by attributes like Name, Description, ImpactOnMentalHealth, and ImpactOnWellBeing. Additionally, the "EEG" class, detailing Model, SensorType, and BrainActivityMonitoring, emphasizes the technological means of monitoring these brain activities. The intricate connections among these classes underscore a network of influence in which video games not

only affect brainwave patterns but also translate into cognitive and emotional effects, which are measurable through EEG technology. The diagram also integrates the "MentalDisorder" and "VRTechnology" classes, representing the broader implications of these interactions in mental health and virtual reality applications, thus painting a multi-dimensional picture of the interconnectedness between gaming, neurological activity, and mental health outcomes.

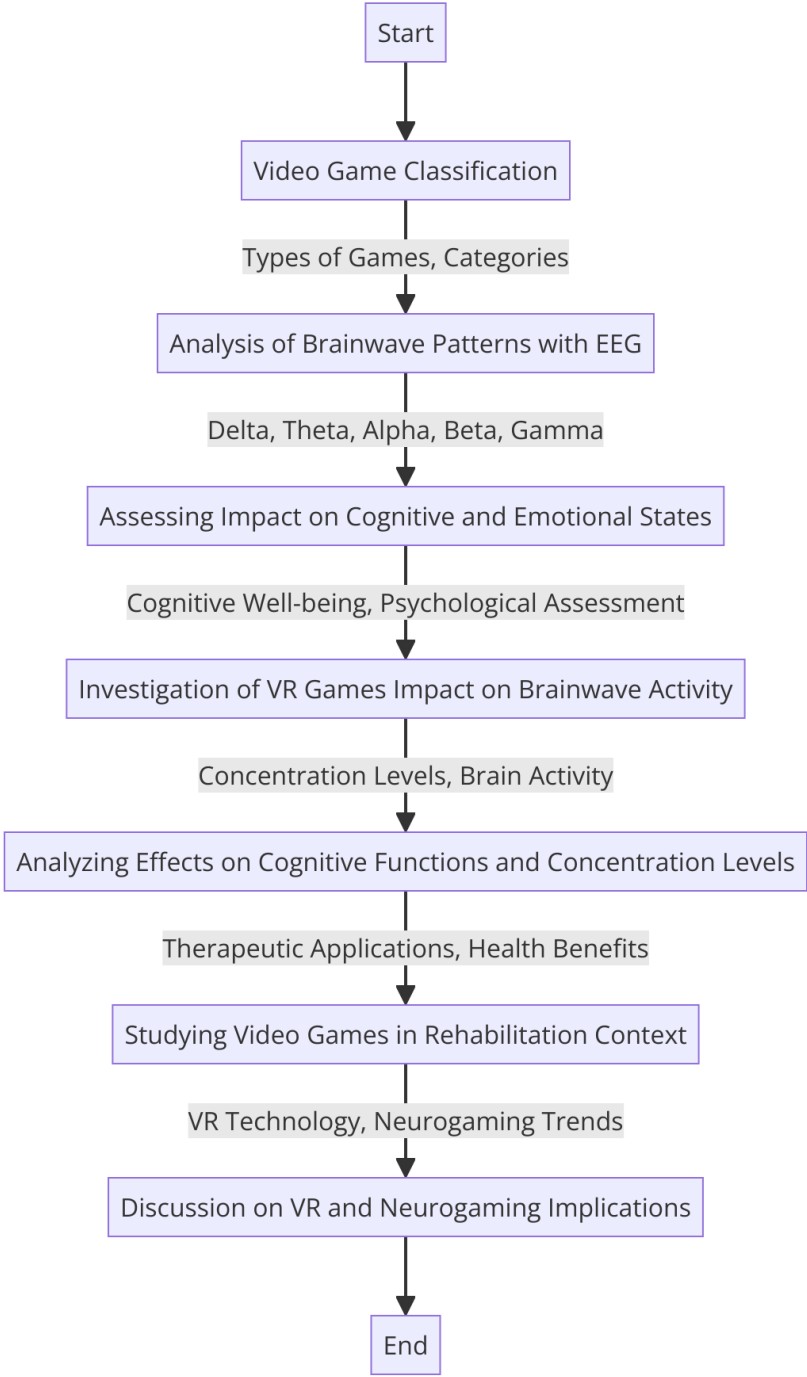

**Figure 7.** Exploring the impact of video games on brain activity: from classification to neurogaming implications.

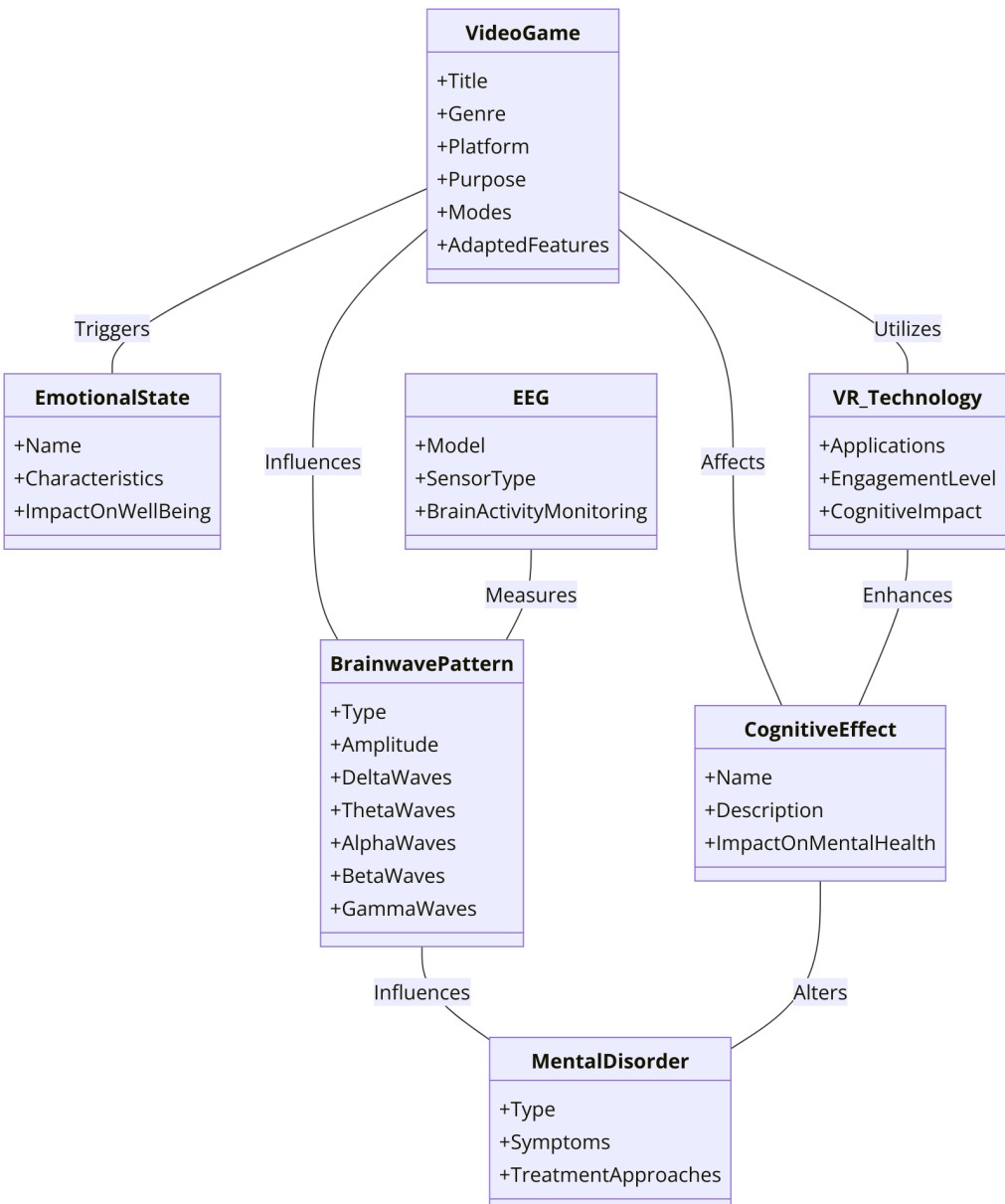

**Figure 8.** Class diagram: interplay of video games, brainwave patterns, cognitive effects, emotional states, EEG monitoring, mental disorders, and VR technology.

The scheme shown in Figure 9 portrays a structured relationship between video games and their multifaceted influence across various domains. At its core, the "VIDEOGAMES" entity encapsulates fundamental attributes like title and genre, serving as a nexus connecting to different research areas. The "LEARNING" entity, linked to "VIDEOGAMES", encompasses studies ranging from Drigas' exploration of online learning in mathematics to Bavelier's analysis of attentional control in action video games, highlighting the diverse educational impacts of video gaming. Further, the diagram branches into "COGNITIVE-EMOTIONAL-STATES", depicting studies like Hughes' investigation of physiological signals in gaming, and "MENTAL-HEALTH", capturing research such as Becker's focus on clinical well-being in schizophrenia patients, illustrating the broad spectrum of cognitive, emotional, and mental health influences of video games. The "MISCELLANEOUS" category encapsulates varied topics like Russoniello's study on mood improvement and stress reduction, underlining the expansive and versatile nature of video games' impact. This ER diagram analytically represents the interconnected nature of video games in educational,

cognitive, emotional, mental health, and various other research fields, underlining their multifarious roles in contemporary research and development.

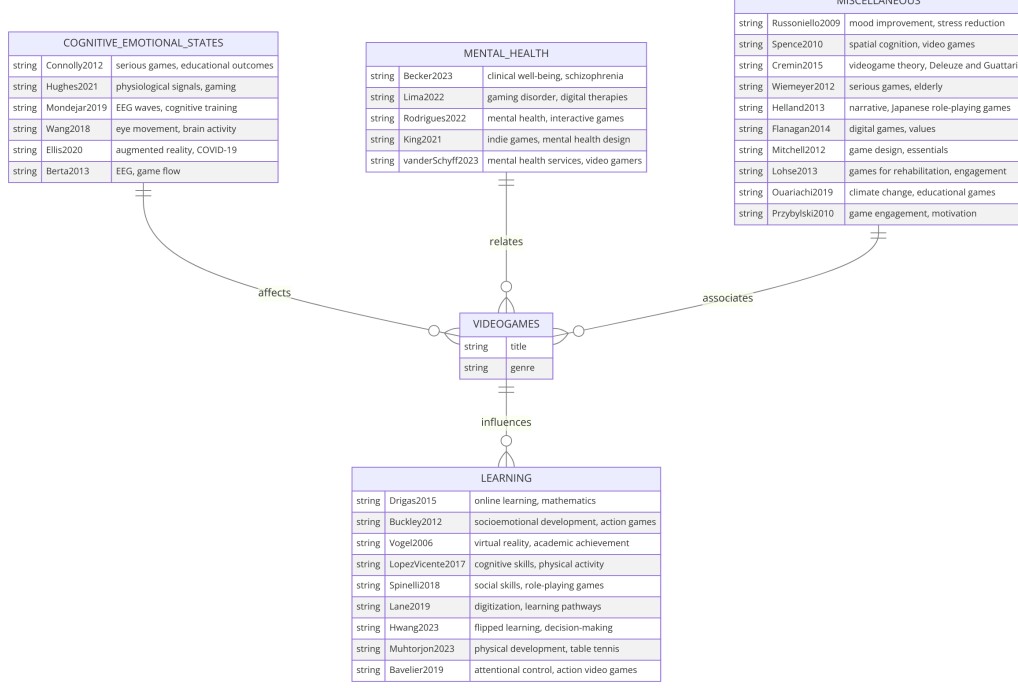

**Figure 9.** Mapping video game research references across learning, cognitive states, mental health. Connolly2012: [12]. Hughes2021: [13]. Mondejar2019: [14]. Wang2018: [15]. Ellis2020: [11]. Berta2013: [10]. Becker2023: [60]. Lima2022: [61]. Rodrigues2022: [62]. King2021: [63]. vanderSchyff2023: [64]. Russoniello2009: [65]. Spence2010: [66]. Cremin2015: [67]. Wiemeyer2012: [68]. Helland2013: [69]. Flanagan2014: [70]. Mitchell2012: [71]. Lohse2013: [72]. Ouariachi2019: [73]. Przybylski2010: [74]. Drigas2015: [41]. Buckley2012: [42]. Vogel2006: [43]. LopezVicente2017: [44]. Spinelli2018: [45]. Lane2019: [46]. Hwang2023: [47]. Muhtorjon2023: [48]. Bavelier2019: [49].

### 3.1. Search Outputs and Results

The statistical parameters associated with the identified articles were as follows: The total of citations from 2000 to 2023 was 14,201. Most of the publications concentrated on the disciplines of computer science (12,461) and engineering (4149), emphasizing video games. Recognizing the interdisciplinary nature of video gaming impacts, we acknowledge the importance of cognitive sciences in providing a holistic view. Therefore, future updates to this review will aim to include key findings from cognitive science and related disciplines to ensure a comprehensive understanding of the cognitive impacts of video games.

In the Scopus database, our review endeavor led us to navigate to the esteemed Document Search section, a pivotal repository of knowledge into which we meticulously inputted the term video game and meticulously appended each of the distinct video game categories delineated in Table 1, thereby enriching the depth and breadth of our investigation. To refine and enhance the precision of our search outcomes, we judiciously employed two discerning filters: firstly, we restricted the temporal scope of our inquiry to the years spanning from 2000 to 2023, thus ensuring a contemporary and relevant dataset; secondly, we discerningly constrained our exploration to the specialized domains of computer science and engineering, thereby focusing our study on the intersection of technology and gaming. The comprehensive details of this search methodology are visually depicted in Figure 10.

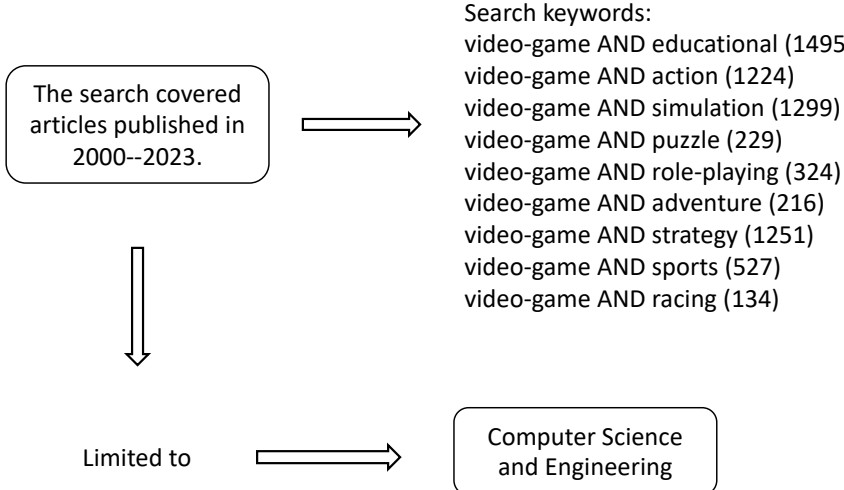

**Figure 10.** Search diagram for video games and their types in the Scopus database.

With a careful examination of the search results, we could discern the intricate relationship between the various keywords utilized, with particular emphasis placed on the initial 5 terms shown in Figure 11. Across all the diverse categories assessed, it became evident that prominent keywords like human–computer interaction, interactive computer graphics, and video games were consistently present. Furthermore, our scrutiny also revealed the presence of other significant keywords, such as humans, virtual reality, and serious games, among others. This collective observation serves to underscore the enduring association between the subject matter of video games and the inherent potential for individuals to derive benefits from such digital entertainment, irrespective of the specific genre involved. It is apparent that the overarching focus remains on fostering meaningful interactions with users and the ultimate aim of enhancing individuals' overall well-being and experiences.

In Figure 12, a detailed description is provided for each of the keywords that are listed within the "game type" column of Table 1, presenting a comprehensive overview of the main types that are emphasized in research papers pertaining to learning, cognitive development, and the multiple variables that have an impact on academic performance. Within this visual representation, spanning from the years 2000 to 2023, there is a clear emphasis on showcasing the number of documents or articles allocated to each specific category, all of which are associated with the term video game. Through this visualization, an insightful analysis can be conducted to observe the evolution of each video game type metric throughout the specified period of study. Outstanding categories such as educational, action, simulation, and strategy emerge as the focal points of scholarly attention during this timeframe in relation to the term "video game". It is evident that the examination of these particular categories is on the rise, which is indicative of a growing interest among researchers, as evidenced by the increasing prevalence of these topics within the realm of academic literature.

Figure 13 exhibits the quantitative data representing the frequency of citations spanning from the years 2000 to 2023, juxtaposing various categories of video games denoted with the term "video game". The analysis differentiates four distinct genres of video games that have garnered considerable scholarly attention: simulation, educational, strategy, and action games. These findings were extracted from the comprehensive repository of academic literature, the Scopus database.

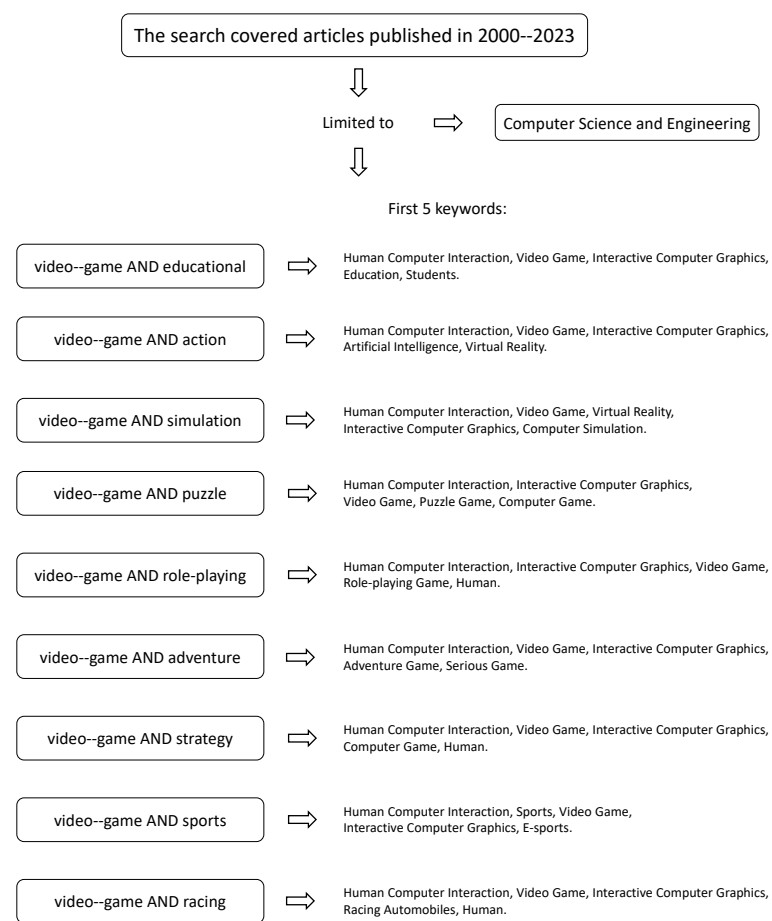

**Figure 11.** Search diagram for video games and the main five keywords.

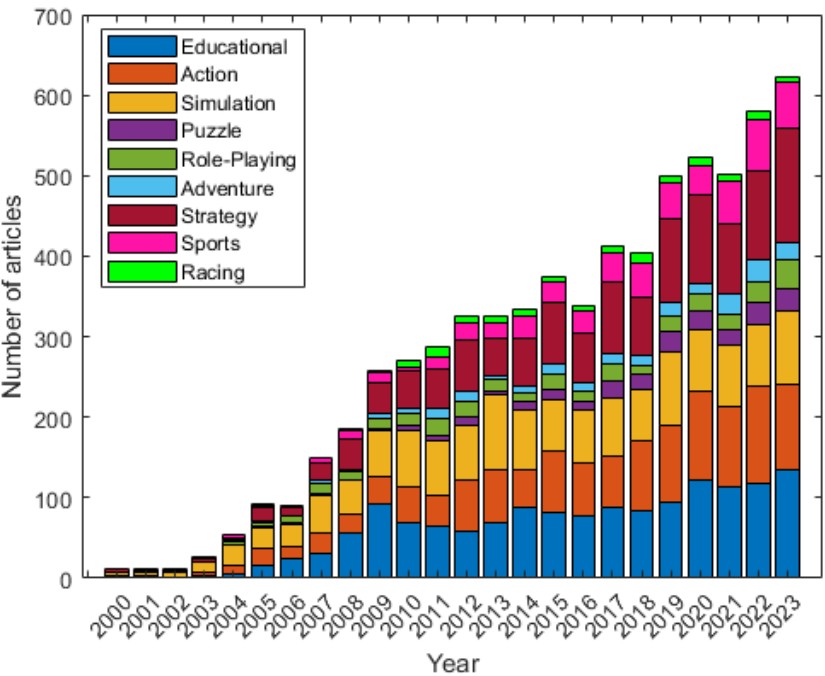

**Figure 12.** The number of research papers pertaining to behavior change techniques that were published from the years 2000 to 2023, as reported in the Scopus database.

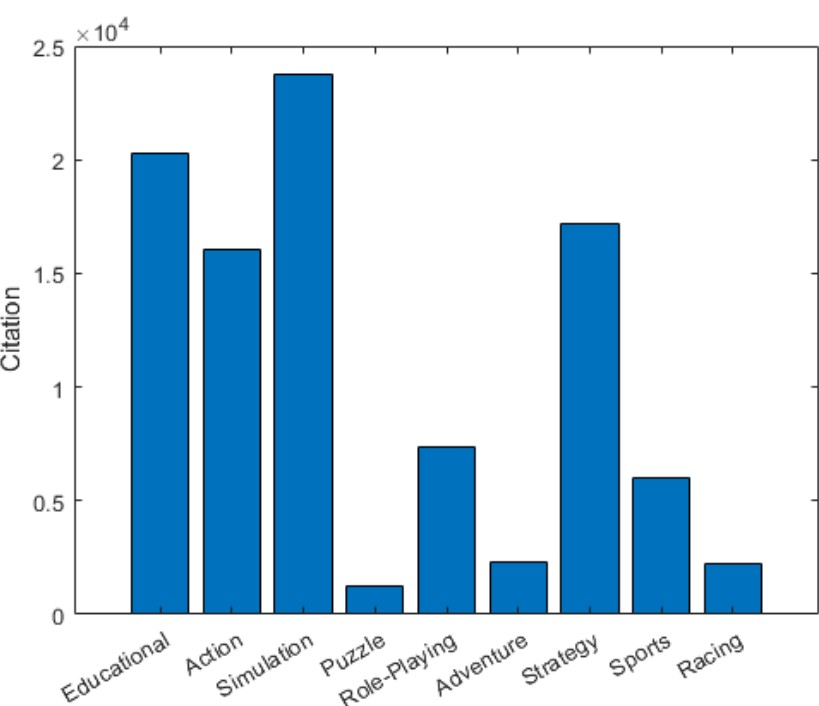

**Figure 13.** Citations for video game type searches from 2000 to 2023 reported in the Scopus database.

### 3.2. Types of Video Games

Passive video games essentially involve the use of hand–eye coordination to interact with an electronic device. These games are generally played in a passive position with minimal body movements. They often receive criticism for promoting sedentary behavior and the consumption of unhealthy foods [75]. On the other hand, active video games are electronic games that allow players to physically interact using their hands, arms, legs, or entire bodies, with images displayed on a screen or output device. Success in these games depends, among other factors, on the movements made in front of a camera, an infrared sensor, a laser, a pressure-sensitive mat, or a modified ergometer. Levac et al. conducted a comprehensive study involving 38 children aged 7 to 12. The objective was to assess the quantity and quality of children's movements while playing boxing and tennis on the Nintendo Wii® (Nintendo in Kyoto, Japan) and skiing and playing soccer on the Wii Fit® (Nintendo in Kyoto, Japan) [76]. They measured the center of pressure displacement as an indicator of the quantity of movement and pelvic movement as an indicator of movement quality using a force platform. Levac et al. found differences only in the quantity of movement, especially for children with prior experience using these devices. Therefore, it can be concluded that there is a learning curve that allows for increased movement as children play. This amount of movement can also be measured to determine energy expenditure, considered an important variable for weight control. There is evidence suggesting that both parents and children believe that active video games promote physical activity and may contribute to reducing childhood obesity [77]. In a focus group study involving seven children and four adults, participants concluded that active video games enhance the amount of physical activity and fitness. However, parents identified barriers such as the cost of video games and limited space at home that hinder their use.

Figure 14 shows three categories of video games and the attributes they evoke, depending on the category of video games. Each video game category can have various characteristic attributes that can make a player feel emotions when playing.

Challenging puzzler games are a type of challenging game that requires players to use their cognitive skills to solve problems. These games can be simple or complex, and they can have a variety of objectives. Puzzle games are typically played on mobile devices, tablets, PCs, consoles, handheld consoles, and virtual reality headsets.

Casual video games are easy to learn and play, and they do not require much time or effort to complete. These games are often fun and relaxing, and they are a popular way to pass the time. Casual games are typically played on mobile devices, tablets, PCs, consoles, handheld consoles, and virtual reality headsets.

Exergames are video games that combine physical activity with fun. They are used to promote physical activity and exercise, and they can be a fun and effective way to get in shape. Exergames are typically played on consoles, handheld consoles, and virtual reality headsets. Exergames, requiring physical movement for participation, are ineffective in increasing levels of physical activity among individuals with various physical conditions such as obesity, heart disease, and diabetes. Moreover, exergames can also play a part in advancing motor skills and coordination among individuals with neurological disorders. Physical therapists might incorporate video games as a tool to support patients with physical ailments in their rehabilitation from injuries and enhance their range of motion, strength, and balance.

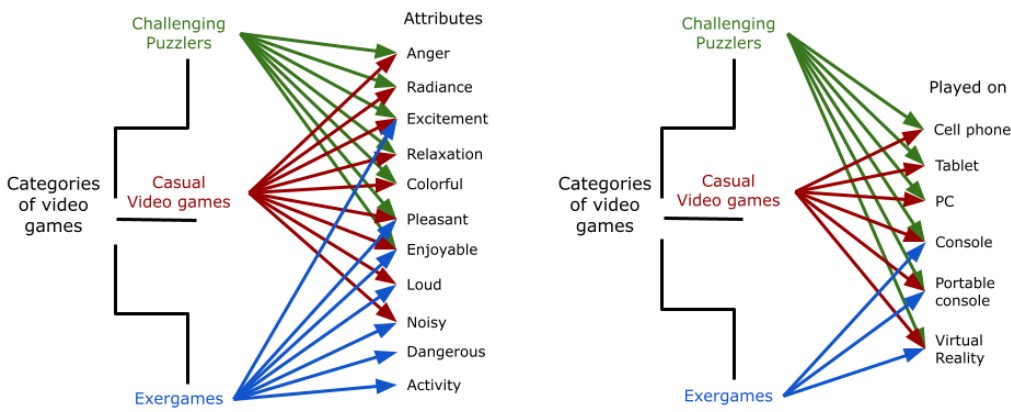

**Figure 14.** Video game attributes and platforms.

### 3.3. Tools for Assessing VR Games

The use of EEGs for assessing task engagement and affective states within the context of neurogaming has gained significant traction, offering researchers a window into the cognitive and emotional aspects of gamers' experiences. EEG headsets, while not medical-grade devices, present an accessible and practical tool for capturing gamers' brainwave signatures [78]. They have been employed to record various aspects of cognitive and affective states, including focused thoughts, creativity, emotional responses to avatars, and the development of affective states. Moreover, they have been compared to more expensive, medical-grade EEG systems, demonstrating their potential for neurogaming applications [79]. Controlled laboratory experiments, as in [80], have validated the efficacy of their indices in diverse contexts, ranging from visualizing shapes to detecting hand movement intentions. This technology has also been used to study cognitive workloads, affective states during film-watching, and dynamic game events, such as character death during gameplay. The findings from these experiments provide valuable insights into the interplay between cognitive and affective processes, fostering the emergence of affective neuroscience in neurogaming research.

One of the primary goals in the field of neurogaming is to establish an optimal relationship among EEG frequency bands, task engagement, and arousal states; yet the standardization of methods remains an ongoing challenge [81]. Researchers have worked toward identifying game-specific classifiers for assessing gaming events and understanding the cognitive and affective states of gamers in real time. Some studies [82,83], mentioned the application of arousal and engagement indices to game-based classifiers utilizing the Emotiv EEG. They filter EEG data into separate frequency bands and employ three classification techniques to assess various levels of gaming experiences [84]. As the neurogaming landscape evolves, there is a growing need to explore new properties, develop game-

specific classifiers, and enhance the understanding of EEGs' potential in enhancing video game design and user experiences.

### 3.4. Video Games Levels of Impact

In the context of video games, different levels of impact can be observed based on the player's experience and engagement. These levels encompass physiological, psychological, sensory–motor, and social aspects.

Physiological level: Video games can have diverse effects on physiological functions, including the cardiovascular, cardiorespiratory, and immune systems, depending on their specific design and purpose. For instance, the work in [65] suggests that gaming might potentially restore the neural plasticity observed during early brain development.

Psychological level: Video games contribute to cognitive experiences and learning processes. Players engage in problem-solving, encounter varying learning scenarios, receive instant feedback, and process background information, thus enhancing cognitive skills. Games also have the potential to positively impact intrinsic motivation, attitudes, self-concept, emotions, perceived control, and self-efficacy. Furthermore, specific components of the gaming experience, such as flow, challenge, tension, and enjoyment, play a pivotal role in the psychological aspects of gaming [66].

Sensory–motor level: The impact on sensory–motor skills and abilities depends on the quality of the human–game interface, the nature of game tasks, and the player's experiences. Video games can lead to the acquisition and transfer of basic or specific sensory–motor skills, including reaction times and balance skills [67].

Social interaction and communication: The significance of social interaction and communication in the context of video games can be highlighted, particularly in massive multiplayer online games. Digital games often provide a platform for players to interact and communicate, fostering a constructivist approach to learning. Additionally, mobile devices like cell phones and personal digital assistants, as well as specific social settings, can be leveraged to facilitate interaction and communication among players [68].

### 3.5. Platform

Platform games are a quintessential genre in the world of video games. In these games, players guide characters through hostile territories to accomplish missions, often involving the rescue of a princess. As they progress, players can collect power-ups and extra lives, making it easier to overcome increasingly challenging obstacles and dangerous adversaries. The game's environments become more intricate as players advance through different screens [69]. Additionally, platform games encompass maze-like experiences and hidden passages in the realm of entertainment software.

Some games blur the line between platform and combat genres; however, the key distinction lies in platform games' emphasis on providing players with the means to navigate obstacles without necessarily destroying them [70]. Players have formidable tools at their disposal to deal with enemies, though.

Platform games have played a pivotal role in introducing labyrinthine challenges and secret passageways. Typically featuring cartoonish and child-friendly characters that can be either human or anthropomorphic animals in the Disney tradition, these games incorporate a wide variety of adversaries, including creatures of all forms—animal, human, or even plant-like [71]. The thematic and visual elements of platform games are generally mild and, as a result, have been a popular choice for developing games centered around beloved children's characters.

### 3.6. Video Games in Rehabilitation

In the context of exploring the relationship between video games and brain activity measured with an electroencephalogram, some works [72,73] discuss the potential of video games in rehabilitation. They emphasize the challenges of patient nonadherence to traditional therapy and propose motion-controlled video games as a promising avenue

to enhance therapy. The articles review evidence supporting the positive effects of video games on cognitive, motor, and affective measures. They advocate for the integration of commercial video games into rehabilitation, utilizing adapted control systems. The discussion revolves around the potential of gameplay to increase patient engagement and motivation, which are crucial factors for successful rehabilitation. It underscores the need for further research to conclusively determine the effectiveness and efficacy of video games in rehabilitation, highlighting the positive impact of well-designed game mechanics and the potential of video games as a supplement to traditional therapy.

The intricate relationship between different types of video games, their physical demands, and the corresponding patterns of brainwave activity is a focal point of this study. By analyzing the amplitude of delta, theta, alpha, beta, and gamma waves during gameplay, the research aimed to elucidate players' cognitive and emotional responses. This investigation, cited in [74], aligns with the growing popularity of video games, surpassing movie theater attendance in the United States. This contributes to scientific understanding by proposing a theory-based motivational model rooted in self-determination theory. This model suggests that video games' appeal and well-being effects stem from their ability to satisfy basic psychological needs for competence, autonomy, and relatedness. The study's exploration of various topics, including need satisfaction, the motivational appeal of violent game content, and the determinants of game engagement underscores the comprehensive approach to understanding video game engagement and its implications for psychological processes and well-being.

Furthermore, video games can be utilized to help patients with physical illnesses learn new motor skills and compensate for lost functionality. They could also be utilized as a method of managing illnesses for individuals with chronic physical conditions. For instance, video games might be utilized to offer patients educational details about their ailments, assist them in keeping track of their symptoms, and provide support and encouragement.

Video games crafted with the intention of augmenting cognitive welfare offer significant prospects for academic enrichment, assistance, and the appraisal of psychological well-being circumstances. These virtual mediums can act as efficacious instruments for the training and aid of medical practitioners, empowering them to advance their expertise beyond the confines of customary therapy. Furthermore, they foster the linkage between individuals and healthcare providers, granting them the means to obtain economical self-assessment and treatment alternatives.

Mental disorders are a significant global health concern, according to the World Health Organization [85]. The WHO recognizes the importance of mental health and emphasizes the need for comprehensive mental healthcare and support. Table 2 describes various mental disorders according to the classification and diagnostic criteria established by the World Health Organization. The descriptions provided in the table serve to encapsulate the multifaceted nature of mental health conditions, offering a nuanced understanding of the wide range of disorders that fall within the realm of psychiatric and psychological discourse.

Within the domain of mental health, serious games have emerged as potent instruments for education and instruction. More specifically, therapeutic games, which constitute a subset of serious games, have been meticulously designed and utilized as supplementary tools to augment psychotherapy for diverse mental disorders [86]. The employment of video games for mental health purposes has generated interest in pioneering approaches to bridge the gap in mental health accessibility, particularly by means of providing readily available, adaptable, and cost-efficient care [87].

Research has given proof that video games have the potential to contribute positively to the treatment of mental disorders through their ability to enhance cognitive functions, alleviate symptoms, and enhance awareness of mental well-being. Various studies have demonstrated that engaging in video game training can result in notable enhancements in cognitive capacities and bring about notable structural and functional changes in the brains of individuals who are considered to be in good health [60]. Moreover, inquiries have suggested that video game training can produce advantageous results for patients

diagnosed with schizophrenia, which include enhancements in the capacity to sustain attention over an extended period of time, reductions in adverse symptoms and general psychopathology, and the perceived recovery of mental health [61]. Furthermore, it is crucial to note that video games could potentially serve as a means of digital therapy for individuals struggling with mental disorders such as depression, anxiety, and attention deficit hyperactivity disorder (ADHD) [62]. Moreover, it is crucial to emphasize that video games that integrate interactive storytelling and decision-making components may promote the awareness of mental health concerns and encourage the adoption of self-care as a preventive measure [63]. By boosting the knowledge and understanding surrounding mental health, video games can contribute to overcoming barriers that hinder individuals from seeking help and can facilitate the overall improvement of access to mental health services [64].

**Table 2.** Mental disorders according to the World Health Organization.

| Mental Disorder | Description |
|---|---|
| Anxiety disorders | Anxiety disorders are characterized by excessive fear and worry and related behavioral disturbances. |
| Depression | Depression is different from usual mood fluctuations and short-lived emotional responses to challenges in everyday life. |
| Bipolar disorder | People with bipolar disorder experience alternating depressive episodes with periods of manic symptoms. |
| Post-traumatic stress disorder (PTSD) | PTSD may develop following exposure to an extremely threatening or horrific event or a series of such events. |
| Schizophrenia | Schizophrenia is characterized by significant impairments in perception and changes in behavior. |
| Eating disorders | Eating disorders involve abnormal eating and preoccupation with food, as well as prominent body weight and shape concerns. |
| Disruptive behavior and dissocial disorders | This disorder is one of two disruptive behavior and dissocial disorders; the other is oppositional defiant disorder. |
| Neurodevelopmental disorders | Neurodevelopmental disorders are behavioral and cognitive disorders that arise during the developmental period and involve significant difficulties in the acquisition and execution of specific intellectual, motor, language, or social functions. |

Furthermore, video games can be utilized skillfully to evaluate and measure cognitive capabilities and psychological conditions. Consequently, this can yield significant information for the advancement of frameworks and forecasts related to mental welfare.

The competition for the largest number of VR customers is getting stronger and stronger, and providing constant innovation in the world of VR is extremely difficult and challenging. This technology is increasingly part of the lives of many people who know how to take advantage of it or simply use it as a source of entertainment.

Table 3 serves as a structured classification of the provided information. It categorizes the topic into four main sections. It provides a clear categorization of the information provided, with distinct sections for adaptive task automation techniques, biometric sensor monitoring, types of game tasks, and automatic difficulty adjustment, along with appropri-

ate references. First, it introduces "Automation Techniques", explaining that they activate automated game task assistance under specific conditions, including identifying a high player workload and maintaining optimal vigilance. Second, it describes "Biometric Sensor Monitoring", highlighting how physiological responses detected via biometric sensors can influence game task sequencing to elicit or suppress player emotions. Third, the table classifies "Types of Game Tasks" into three distinct categories: "Explicit Tasks", which are presented objectives, goals, and missions; "Implicit Tasks", unspoken but expected goals like staying alive or maximizing skills; and "Player-Driven Tasks", created through player creativity within game mechanics, often leading to emergent gameplay. Finally, it mentions "Automatic Difficulty Adjustment", explaining that game difficulty levels can be automatically modified for each type of game task. This organized categorization provides a comprehensive overview of adaptive techniques in gaming, making it easier to understand the subject matter.

**Table 3.** Categorization of adaptive task automation techniques in games.

| Category | Description |
| --- | --- |
| Automation techniques | Adaptive task automation techniques initiate automated assistance for game tasks under specific conditions: when a high player workload is detected or to maintain optimal vigilance during task execution. |
| Biometric sensor monitoring | Game tasks' sequencing in gameplay can be monitored based on physiological responses measured via biometric sensors. This monitoring aims to elicit or suppress specific player emotions. |
| Types of game tasks | Game tasks can be broadly classified into three categories. |
| Explicit tasks | Tasks explicitly presented to the player as part of the gameplay. These may include objectives, goals, and missions. |
| Implicit tasks | Tasks that are not explicitly stated in the game interface but are expected to be fulfilled. Examples include "stay alive", "maximize your skills", or "collect as many items as possible". |
| Player-driven tasks | Tasks created through the player's creativity within the limitations of the game mechanics. These lead to emergent gameplay, which is typical of games like Minecraft™ and those with no predefined narratives. |
| Automatic difficulty adjustment | Game difficulty levels can be automatically adjusted for each of the three types of tasks. |

Table 4 concisely categorizes video games based on their characteristics, specifically focused on game purpose, mode, and adapted game features. This classification aligns with the accompanying text that discusses a comparison of affective video game properties. It reveals that the majority of affective games primarily serve the purpose of entertainment, with a few exceptions in the field of applied gaming related to team cognition, communication, coordination, and clinical studies of concentration levels. These games vary in genre, including arcade, puzzle, FPS, platform, and car-racing games, and most are desktop applications. The preferred mode for experimental games is single-player, although there are instances of two-player and multiple-player modes. The table also demonstrates that the adaptation of game features extends beyond dynamic difficulty adjustment (DDA) to encompass additional aspects such as audio-visual properties, environmental factors,

territorial control, tunnel vision effects, visibility, and the adjustment of the skills and properties of non-player characters and opponents. The table effectively categorizes affective video games' diversity and associated purposes, modes, and adapted game features, as highlighted in the provided text.

Figure 15 illustrates the number of records linked to mental disorders as documented by the World Health Organization. The chart's horizontal axis enumerates diverse mental disorders, whereas the vertical axis depicts the number of documents. It is crucial to emphasize that the quantity of documents illustrated in the chart does not necessarily signify the prevalence of a psychological disorder. Nevertheless, it is discernible that there is a higher frequency of documents pertaining to depression, owing to its status as a significant public health concern. Depression impacts a substantial number of individuals globally and stands as the primary cause of disability resulting from mental health issues [88]. Investigating depression is imperative to enhancing public health outcomes, as well as advancing treatment methodologies and preventing the emergence of the disorder.

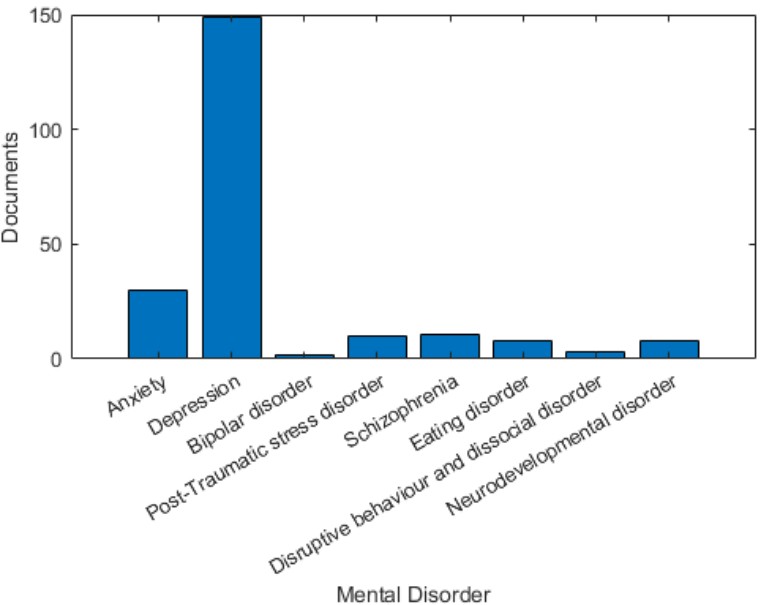

**Figure 15.** Number of publications in the Scopus database on mental disorders (2000–2023), each related to video games.

Table 5 presented below provides a comprehensive summary of various scholarly research papers focusing on the impact of video games on individuals' mental health. By thoroughly scrutinizing these studies, video games' consequences on cognitive functions, emotional well-being, and overall mental health are evaluated and merged. The columns in the table contain the following information: work, the title of the research paper; analysis, which describes the analysis method used in the paper; video game method, which describes how the video game was used in the study; measurements, which indicates what was measured and studied in each research; N, the number of participants in the study; treatment conditions, which describes the different conditions that participants were assigned to; main findings, a summary of the most important results of the study; and limitations, which describes the study's limitations, such as its sample size or study design. The presented table gives a thorough and detailed overview of the ongoing studies in the domain of video games and their impact on mental wellness. The extensive body of studies indicates that video games possess the potential to yield favorable outcomes for mental health.

Furthermore, it is important to note that the subsequent Table 6 encapsulates a detailed analysis of findings that have been extrapolated from various studies carried out within the timeframe of the past five years. These studies have specifically shed light on the significant

potential that video games hold in terms of assisting individuals who are grappling with various mental health conditions. The overarching aim of presenting this table is to offer a thorough and exhaustive examination of the contemporary uses of video games within therapeutic contexts. This involves placing a strong emphasis on the myriad ways in which video games can positively impact the overall well-being and cognitive functions of patients who are undergoing treatment.

**Table 4.** Classification of affective video games.

| Reference | Game Purpose | Mode | Adapted Game Feature |
|---|---|---|---|
| [89] | Entertainment | Single-player | Dynamic difficulty adjustment (DDA) |
| [90] | Applied (team cognition, communication, and coordination) | Single-player | Dynamic difficulty adjustment (DDA), audio-visual properties |
| [91] | Applied (clinical studies of concentration level) | Single-/multiplayer | Dynamic difficulty adjustment (DDA), audio-visual properties |
| [92] | Entertainment | Single-player | Dynamic difficulty adjustment (DDA), audio-visual properties, environmental density and gravity |
| [93] | Entertainment | Single-/multiplayer | Dynamic difficulty adjustment (DDA), skills of non-player characters (NPCs), environmental properties |
| [89] | Entertainment | Multiplayer | Dynamic difficulty adjustment (DDA), enemy behavior and properties |
| [89] | Entertainment | Single-player | Dynamic difficulty adjustment (DDA), enemy spawn, health, weapon control, boss appearances |
| [94] | Entertainment | Single-player | Dynamic difficulty adjustment (DDA), audio-visual properties, territorial control, tunnel vision effects, visibility |
| [95] | Entertainment | Single-player | Dynamic difficulty adjustment (DDA), enemy spawn, health, weapon control, boss appearances |
| [96] | Entertainment | Single-player | Dynamic difficulty adjustment (DDA), audio-visual properties, territorial control, tunnel vision effects, visibility |

**Table 5.** Works and citations.

| Work | Analysis | Video Game Method | Measurements | N | Treatment Conditions | Main Findings | Limitations |
|---|---|---|---|---|---|---|---|
| [97] | Diagnosing gaming disorder without pathologizing healthy behavior | Critical examination comparison | Previous research conceptualization | NA | Non-problematic gaming patterns of gaming disorder | Distinguish between high involvement and pathological involvement in video games | The lack of evidence for functional impairment in many emerging behavioral addictions and the mixing of core and peripheral criteria in studies |
| [98] | Relationship between video game play time and subjective well-being | | | 6011 | | | |
| [99] | Examines the video game therapy methodology, which employs video games as a means of therapeutic intervention for mental health ailments | Analyzes the use of video games in therapy across three associations, focusing on emotional, social, and cognitive aspects | Myers–Briggs Type Indicator (MBTI) | 5 | Depression, anxiety, PTSD, and addiction | Video games can effectively treat mental health conditions like depression, anxiety, PTSD, and addiction | The impact of violent video games varies greatly among individuals, which can affect the therapy's effectiveness |
| [100] | Games are emerging as tools for mental health education, support, and assessment | Scoped review of video games and VR for assessing anxiety and depression | Gaming effects on pain-related fear and mental health outcomes measured using meta-analysis in chronic pain patients | NA | Schizophrenia, anxious symptoms, physical and psychological health, anxiety, and depression | Video games and VR show potential as assessment tools for anxiety and depression | The potential for risks and threats to players' safety and/or well-being is not fully explored; difficulty in maintaining long-term engagement with serious games |

**Table 5.** *Cont.*

| Work | Analysis | Video Game Method | Measurements | N | Treatment Conditions | Main Findings | Limitations |
|---|---|---|---|---|---|---|---|
| [101] | Combines various psychotherapy approaches with gaming for a more appealing treatment | Employs cognitive behavioral techniques adapted to the context of video games | Focuses on self-regulation, anxiety, and autism spectrum disorders | NA | Utilizes narrative, collaborative, cognitive behavioral, and other evidence-based approaches in psychotherapy | Gaming metaphors and interactive media can be integrated with various therapeutic approaches | The effectiveness of video game-based therapy may vary, depending on the specific mental health condition |
| [102] | Challenges the traditional view of video games by highlighting their potential positive effects on stress reduction, social connection, and cognitive enhancement | Use of commercial video games for therapeutic purposes in mental health | Commercial video games could be incorporated into rapport-building and social skill-training therapy | NA | Stress and daily hassles | Video games enhance a range of cognitive skills | The effects of video games can vary widely among individuals, making it difficult to generalize findings |
| [103] | Development of therapeutic games may focus on genres like music, role-playing, and survival horror for mental health treatment | Conducted an anonymous cross-sectional survey | Collected data on play style, genre, and perception of psychological impact | 2107 | Emotional well-being | 88.4% of participants reported emotional benefits from gaming | The cross-sectional design precludes causal inferences |
| [104] | Identifies adventure and role-playing games as having associations with lower anxiety, depression, and neuroticism | Asks participants about their favorite video game in conjunction with mental health and personality assessments | Utilizes a correlational design to analyze survey data | 200 | Anxiety, depression, and neuroticism | Adventure and role-playing video games are associated with lower levels of anxiety and depression | The study uses a correlational design, which does not establish causality between video games and mental health outcomes |

**Table 5.** *Cont.*

| Work | Analysis | Video Game Method | Measurements | N | Treatment Conditions | Main Findings | Limitations |
|---|---|---|---|---|---|---|---|
| [105] | Challenges the notion that raw playtime is a significant predictor of mental health in gamers | Assists in the identification of at-risk individuals for internet gaming disorder by focusing on social and motivational factors | Conducts an exploratory regression analysis to determine the predictive power of gaming motives and social context on mental health | 13,464 | Provides evidence that could help in identifying at-risk groups for internet gaming disorder | Raw playtime is not a significant predictor of mental health among gamers | Examines the long-term mental health effects of different gaming patterns and contexts |
| [106] | Explores video game play's role in mental and behavioral health recovery among military veterans | Employs thematic analysis to identify patterns in veterans' gaming habits and their impact on mental health | Conducts semi-structured interviews with military veterans | 20 | Veterans were in treatment for mental or behavioral health problems | Gaming helps with mood management and stress relief and provides adaptive coping mechanisms | Gender differences in gaming experiences and preferences may require targeted interventions |
| [107] | Investigates the effects of video gaming on cognitive functions in schizophrenia patients | Cognitive tests for processing speed, attention, working memory, and problem-solving | EEG and MRI to measure brain activity and structure | 234 | Schizophrenia | The study aims to assess working memory function and other cognitive and social functions as primary and secondary outcomes | Reliance on self-reported gaming diaries to verify intervention fidelity |
| [108] | The study aims to see whether video game training could improve cognition in older adults | Tests are conducted before and after training to assess working memory and selective attention | Tests are conducted before and after training to assess working memory and selective attention | 75 | Improvement, motivation, and engagement | Older adults improved in the video games they practiced during the training sessions | The number of participants might not be enough to show strong results, and more participants could help find clearer effects |
| [109] | How video games affect individuals with a persistent low mood | Researchers conduct a reflexive thematic analysis of the interview data | The study uses semi-structured interviews to collect data from participants | 18 | Low mood and depression | Gaming gave participants a sense of achievement and success | The qualitative nature of the study means that the results are subjective and may not be easily replicated |

**Table 5.** *Cont.*

| Work | Analysis | Video Game Method | Measurements | N | Treatment Conditions | Main Findings | Limitations |
|---|---|---|---|---|---|---|---|
| [110] | It aims to understand how therapists use the game and identify factors that affect its use | Thematic analysis is used to identify key themes from the interview and survey data | An online survey is conducted to gather data from therapists | 95 | The game supports cognitive behavioral therapy (CBT) for children with anxiety or a low mood | Young people generally liked using the game as part of their therapy | The diversity of the participants, like their ages, where they are from, or how serious their mental health issues are, might not be wide enough |
| [111] | Investigates the impact of video game play on well-being | Examines the role of players' motivations on their well-being | Uses self-reported well-being surveys from players | 38,935 | Observational study of video game play and well-being | Video games have a very small effect on players' well-being | The study relies on self-reported data, which can be biased or inaccurate |
| [112] | Impact of puzzle games on stress and cognitive functions | Participants are divided into control and experimental groups to observe the effects of watching versus playing a puzzle game | Salivary biomarkers (cortisol and alpha-amylase), electroencephalography (EEG), and Paced Auditory Serial Addition Test (PASAT) | 44 | Stress and cognition indicators | The study suggests that puzzle games could be used as a form of positive cognitive therapy | The effects of the puzzle game are only tested on a single game, which may not represent all puzzle games |
| [113] | Hormones and neurotransmitters play a key role in mental and emotional states and self-awareness | The study looks at how digital games affect hormones and brain changes | The authors focus on studies published after 2010 but also include some older ones if they are relevant | 8 (male) | Psychiatric or long-term degenerative disorders | Digital games can change hormone levels, which affect mood and decision-making | Only one study found concerning the effect of games on dopamine levels |
| [114] | Relationship between video game play and well-being in individuals with first-episode psychosis (FEP) | Surveys 88 individuals with first-episode psychosis (FEP), 57 of whom played video games and 31 did not | Participants complete a range of questionnaires related to video game play and well-being | 134 | First-episode psychosis (FEP) | People with first-episode psychosis (FEP) who played video games report better well-being than those who did not play games | The study does not provide specific details on the limitations of the study |

**Table 5.** *Cont.*

| Work | Analysis | Video Game Method | Measurements | N | Treatment Conditions | Main Findings | Limitations |
|---|---|---|---|---|---|---|---|
| [64] | Perspectives of male video game players on improving access to mental health services | A qualitative question asks participants for their thoughts on better-supporting access to mental health services | The study analyzes responses from the survey, focusing on the qualitative text-based data | 2515 | The study does not specify particular treatment conditions for mental health issues | Comfortable settings are important for effective mental health treatment | The study does not include the perspectives of mental health clinicians |
| [115] | The study focuses on video game disorder (VGD) among university students | Information on weekly gaming hours, daily sleep hours, favorite game types, and reasons for playing is gathered | A cross-sectional study design is used to collect data from university students | 2364 | Video game disorder | Students played games mostly to improve their game character, relax, and have fun | The study uses a convenience sample, which may not represent all university students |
| [116] | How gamification can support mental health using game design elements | In-depth interviews are conducted to gather data | Narrative inquiry is the specific qualitative method applied | 5 | Anxiety issues | Engagement is highlighted as the most significant element, enhancing passion and emotional involvement | The study uses a qualitative approach, which may not capture the full range of experiences or be generalizable |

**Table 6.** Mental disorders and the use of video games.

| Work | Mental Disorder | Type of Video Game | Sample Size | Date Published | Citations | Conclusions |
|---|---|---|---|---|---|---|
| [117] | Anxiety | Using video games to identify components that reduced anxiety through on players' breathing | 67 | 10 June 2022 | 3 | The video game helps people feel calmer and less anxious |
| [118] | Anxiety | How people with social anxiety interact with a computer character | 191 | 5 May 2021 | 26 | Video games could be a way to find out who has social anxiety early on |

**Table 6.** *Cont.*

| Work | Mental Disorder | Type of Video Game | Sample Size | Date Published | Citations | Conclusions |
|---|---|---|---|---|---|---|
| [119] | Anxiety | Designing a video game to study its impact on anxiety reduction | 15 | 21 August 2022 | 2 | Therapeutic video games can help reduce anxiety |
| [120] | Depression | The positive impacts of music-themed casual video games | 56 | 2 August 2022 | 4 | Music-based video games can significantly reduce feelings of depression |
| [121] | Depression | The game seems to help the body's stress system balance better | 61 | 17 September 2019 | 36 | A casual video game can reduce symptoms of treatment-resistant depression |
| [122] | Depression | Survey people who engage in video gaming and belong to a gaming group | 265 | 31 October 2022 | 1 | Gamers feel that playing video games helped their social skills |
| [123] | Bipolar disorder | Improve cognitive outcomes, clinical and skill generalization | 50 | 28 February 2023 | 3 | Virtual reality has potential advantages for video-game-like interventions |
| [124] | Bipolar disorder | Serious games to fight mental health stigma | 313 | 14 February 2022 | 4 | The game was extremely good at decreasing fear related to mental health issues |
| [125] | Post-traumatic stress disorder | Tetris could be a helpful extra treatment for PTSD | 20 | 30 June 2020 | 41 | Tetris could serve as a beneficial supplementary treatment for people with PTSD |
| [126] | Post-traumatic stress disorder | A game to identify support much earlier than traditional methods | 49 | 31 August 2021 | 4 | A score in the game could be indicative of probable PTSD |

**Table 6.** *Cont.*

| Work | Mental Disorder | Type of Video Game | Sample Size | Date Published | Citations | Conclusions |
|---|---|---|---|---|---|---|
| [127] | Schizophrenia | Involved two main types of training for people with schizophrenia | 25 | 29 June 2022 | 2 | Video games offer additional benefits for people with schizophrenia |
| [128] | Schizophrenia | Compare the effects of HIIT and AVG on neurocognition in patients | 82 | 28 February 2021 | 23 | Both types of exercise were beneficial for people with schizophrenia |
| [129] | Schizophrenia | Mediation models to explore whether patients have internet gaming disorder | 104 | 14 January 2021 | 30 | Online gaming helped manage stress and negative feelings |
| [130] | Eating disorder | Tool designed to help young women and girls at risk for eating disorders | 58 | 31 May 2022 | 6 | Majority of participants found it to be a positive experience |
| [131] | Eating disorder | Compare the effects of two groups at high risk for eating disorders | 92 | 17 February 2022 | 1 | The gamified program helped participants to feel better |
| [132] | Disruptive behavior and dissocial disorder | Focuses on the social interactions playing a video game | 3 | 31 January 2019 | 71 | Online multiplayer games are found to support social interactions |
| [133] | Disruptive behavior and dissocial disorder | Video game habits linked to aggressive behavior or trouble in social situations | 111 | 11 July 2022 | 4 | Playing games does not affect these behaviors |
| [134] | Disruptive behavior and dissocial disorder | Collaborative games to improve communication skills | 24 | 18 November 2020 | 11 | Suggest that VR might offer a more effective medium for communication abilities |

**Table 6.** *Cont.*

| Work | Mental Disorder | Type of Video Game | Sample Size | Date Published | Citations | Conclusions |
|---|---|---|---|---|---|---|
| [135] | Neurodevelopmental disorder | Test including physical activity, diet, screen time, and sleep habits | 23 | 13 May 2021 | 4 | Participants showed an increase in physical activity and sleep duration |
| [136] | Neurodevelopmental disorder | Tool designed to detect attention-related problems and impulsive behavior | 103 | 28 October 2022 | 5 | Identifies signs of attention-related issues and impulsive behavior |
| [137] | Neurodevelopmental disorder | Ensuring the application meets their specific needs and preferences. | 27 | 22 September 2019 | 19 | Help people learn better by combining digital and physical tools |

## 4. Results

According to neurofeedback [138], neurons that communicate with each other through electrical pulses that can be measured are called brain waves. These waves have different types of frequencies, and in order to see them, an EGG is needed to record the brain's electrical activity. Brain waves are divided into distinct types, each associated with different states of consciousness, mental activities, and emotional states. Some of the most well-known types are as follows:

- Delta waves, which represent the lowest-frequency waves in an EGG and are mainly associated with when you are relaxed or in a deep sleep.
- Theta waves, which have a slightly higher frequency and predominate when the senses are processing internal information; they occur during deep meditations and are of great importance during learning and memory.
- Alpha waves, which occur when the body is relaxed but, at the same time, active at any time. These help with mental coordination, calmness, and alertness.
- Beta waves, a frequency that is fast, present when we are attentive, and involved in solving everyday tasks or problems, as well as during decision-making or when we are concentrating.
- Gamma waves, which are the fastest, with shorter bursts, and are associated with higher cognition, sensory perception, and consciousness. They are related to when there is a simultaneous information process in several areas of the central nervous system.

Furthermore, it is crucial to mention that brain waves manifest a multitude of groupings based on their particular frequencies, which are quantified in Hertz (Hz). Within the realm of neuroscience, Table 7 illustrates the categorization of brain waves according to their respective hertz measurements.

**Table 7.** Description of brain waves.

| Waves | Frequency Range |
|-------|-----------------|
| Delta | 0.2–4 Hz |
| Theta | 4–8 Hz |
| Alpha | 8–12 Hz |
| Beta | 12–30 Hz |
| Gamma | 30–90 Hz |

The study of the brain and how it works is essential for companies that develop video games since they need to be constantly innovative to provide the best experiences that are more immersive every time. Knowledge about brain waves answers different visual and auditory elements in a VR environment to optimize the user experience. At the hardware level, brain waves are used to control devices, allowing users to interact more and more realistically with the digital world. Some video games are specifically developed for therapeutic purposes or to provide better health for the user, such as stress or anxiety in many cases.

Brain waves are associated with the following brain activities.

- Delta: deep sleep and deep relaxation
- Theta: relaxation, meditation, and light sleep
- Alpha: relaxation, concentration, and mindfulness
- Beta: wakefulness, concentration, and cognitive processing
- Gamma: learning, memory, and complex information-processing

In the realm of brainwave measurement, tools like the MUSE 2 play a pivotal role. The MUSE 2 is one of the sophisticated devices available for measuring these brain waves across the delta, theta, alpha, beta, and gamma frequencies. It utilizes four sensors strategically positioned to capture brain activity from different regions. These sensors, located on the

forehead, behind the ears, and at the back of the head, are instrumental in providing a comprehensive picture of brain function. There are four sensors used with the MUSE 2 (Figure 16) to measure brain activity in the delta, theta, alpha, beta, and gamma frequencies. These sensors are located on the forehead, behind the ears, and at the back of the head. TP9 refers to the brain activity located on the forehead, just above the left eyebrow. AF7 refers to the brain activity on the forehead, just above the right eyebrow. AF8 refers to the brain activity located at the back of the head, just above the right ear. TP10 refers to the brain activity located at the back of the head, just above the left ear.

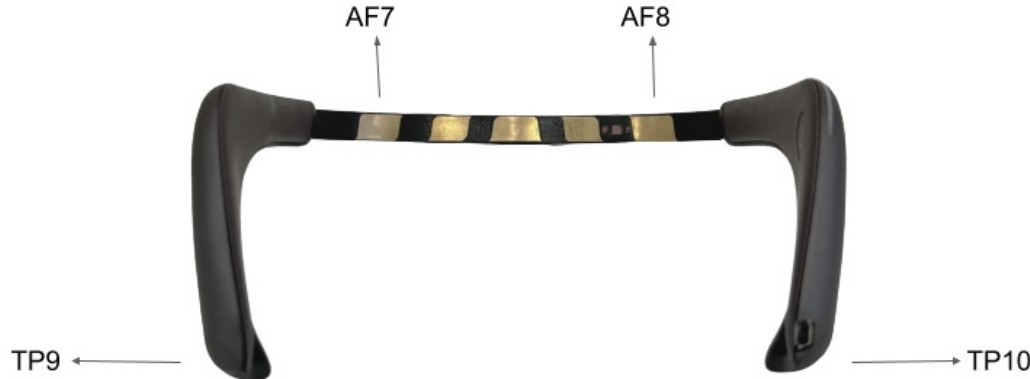

**Figure 16.** Sensors of the MUSE 2.

## 5. Discussion

Virtual reality technology has made remarkable strides and continues to reshape the landscape of digital entertainment. Its unique ability to provide users with an immersive and distraction-free experience, while demanding heightened physical engagement and active decision-making, has marked a significant departure from conventional forms of entertainment. This transformative technology's impact extends across various facets of our lives, with the video game industry emerging as a pioneering force in this journey. Beyond the realm of entertainment, the integration of VR holds the promise of revolutionizing our interactions with both the physical and digital worlds, making it an intriguing subject of exploration and research. The intersection of neurogaming and virtual reality offers a promising avenue for the assessment of cognitive and emotional states in video gamers. As VR gaming continues to evolve and engage players in immersive experiences, there is a growing need to understand how these experiences influence the human mind. The integration of EEG-based BCIs into gaming environments provides a unique opportunity to gain real-time insights into user states during gameplay. This knowledge is invaluable to the development and assessment of video game design. EEG data, collected continuously and unobtrusively, can offer objective measures of cognitive workload, stress levels, and task engagement, among other factors. This approach provides a comprehensive and quantitative understanding of the gamer's mental and emotional states, allowing researchers to detect variations and sources of these variances as individuals interact with VR games.

The analysis of brainwave patterns in response to various virtual reality games represents a crucial aspect of understanding the profound impact that VR gaming can have on players. The measurement of delta, theta, alpha, beta, and gamma waves offers a window into the intricate interplay between gameplay and cognitive responses. This research delves into the complexities of VR gaming by investigating the relationships and patterns in concentration and attention levels exhibited by individuals while engaging with these immersive experiences. By focusing on the data generated via a VR device, particularly delta, theta, alpha, beta, and gamma waves, it becomes possible to gain deeper insights into how different games influence the cognitive and emotional states of players.

The selection of these specific brainwave indicators, delta, theta, alpha, beta, and gamma, is significant for understanding the cognitive aspects of VR gaming. These indicators serve as valuable markers of a player's mental and emotional state during gameplay,

allowing for a comprehensive analysis of how individuals respond to video games in the virtual reality environment. By examining the variations in these brainwave patterns across different game types, we can uncover nuanced insights into the cognitive and emotional dimensions of the gaming experience. This comprehensive approach to analyzing the interplay between VR games and brainwave activity offers a unique perspective on the potential impact of these games and enriches our understanding of the immersive gaming landscape.

The measurement of delta, theta, alpha, beta, and gamma waves during gameplay represents a crucial advancement in our understanding of the impact of video games on the physical and emotional behavior of individuals. This comprehensive analysis sheds light on the intricate interplay between different game genres, their associated physical demands, and the corresponding patterns of brainwave activity. By dissecting the cognitive and emotional responses through neurophysiological markers, this study significantly contributes to the existing body of knowledge. The identification and interpretation of specific brainwave patterns provide nuanced insights into the immediate and enduring effects of video game engagement. Such investigations extend the current state of the art by elucidating the positive influence of video games on the psychological conditions of individuals. This research not only enhances our comprehension of the underlying mechanisms but also underscores the potential for tailored game experiences to positively impact mental well-being, offering valuable implications for the design of therapeutic interventions and the promotion of positive psychological outcomes through gaming.

One of the intriguing aspects of virtual reality is its profound impact on the human brain. The analysis of brainwave patterns, including delta, theta, alpha, beta, and gamma waves, in response to various virtual reality games provides valuable insights. Different game types have been found to exert varying influences on brain wave activity.

## 6. Conclusions

Virtual reality technology continues to advance and revolutionize digital entertainment. It offers users a more immersive and focused experience by eliminating real-world distractions and demanding increased physical engagement and active decision-making. VR technology is significantly impacting various aspects of our lives, with the video game industry leading the way. As VR integration expands beyond entertainment, it offers a potential revolution in how we interact with both the real and digital worlds. Virtual reality technology underwent significant growth, with a 92.1% increase in 2022, notably driven by the success of Meta glasses. This technology is extending its reach beyond entertainment, with applications in various industries, from work tools to healthcare. Therefore, while the existing analysis provides a robust overview of VR's trajectory through 2022, it is crucial to acknowledge that the 92% growth rate might indeed be an underestimate of the current market dynamics. Preliminary estimates suggest that the growth rate for VR in 2023 surpasses 95%, reflecting continued innovation and market penetration. As comprehensive industry reports for 2023 become fully available and are subsequently analyzed, they are expected to confirm this trend of accelerated growth in the adoption and technological refinement of VR. This anticipated increase is particularly significant in discussions concerning investment in VR technology and the strategic planning for businesses operating within this burgeoning sector.

The analysis of delta, theta, alpha, beta, and gamma waves in response to different virtual reality games demonstrates that each game type can significantly influence brain wave activity. The consistent activity of gamma waves across different VR games can be attributed to the standardization of sensory stimuli, particularly in the visual and auditory aspects. VR headsets and headphones create a consistent sensory focus, resulting in a consistent level of cognitive load across games. By employing electroencephalogram technology, this research offers a groundbreaking exploration of how video games affect cognitive and emotional states. The meticulous analysis of various brainwave patterns during gameplay in VR environments yields vital insights into how different gaming experi-

ences impact mental processes. This study bridges the gap between gaming, psychological, and neurological studies, establishing a new benchmark for similar research in this domain.

Moreover, the implications of this research in mental health are profound. The study elucidates the potential of video games, particularly in VR settings, as a therapeutic tool. By examining the effect of various game genres on brainwave responses, it opens avenues for developing tailored video game therapies for mental healthcare. This approach suggests a significant shift in how we understand and use entertainment technologies for health benefits, highlighting the transformative potential of gaming in enhancing cognitive processes and mental well-being. Thus, this research not only contributes to the theoretical understanding of neurogaming but also paves the way for practical applications in healthcare and education, marking a significant step forward in interdisciplinary research.

This research represents a significant intersection between technology and medicine. The study provides groundbreaking insights into the neurological impacts of gaming. This research contributes significantly to both fields by demonstrating the therapeutic potential of video games in mental healthcare. It highlights the capacity of gaming technologies to not only entertain but also facilitate cognitive and emotional regulation, thus offering innovative approaches in therapeutic settings. Consequently, this work not only furthers our understanding of neurogaming but also establishes a new frontier in medical technology, where gaming can be harnessed as a powerful tool in mental health treatment and cognitive rehabilitation.

**Author Contributions:** Conceptualization, J.G.-B., C.D.-V.-S., J.A.D.-P.-F. and R.A.B.; methodology, J.G.-B., C.D.-V.-S. and R.A.B.; validation, J.G.-B., C.D.-V.-S. and R.A.B.; formal analysis, J.G.-B., C.D.-V.-S. and R.A.B.; investigation, J.G.-B., C.D.-V.-S., J.A.D.-P.-F. and R.A.B.; resources, J.G.-B., C.D.-V.-S. and R.A.B.; writing—original draft preparation, J.G.-B., C.D.-V.-S. and R.A.B.; writing—review and editing, J.G.-B., C.D.-V.-S. and J.A.D.-P.-F.; supervision, C.D.-V.-S. and J.A.D.-P.-F.; project administration, C.D.-V.-S.; funding acquisition, J.G.-B. and J.V.-A. depicted some formal concepts and reviewed the manuscript. All authors have read and agreed to the published version of the manuscript.

**Funding:** This research received no external funding.

**Institutional Review Board Statement:** The Integrity Code of the Universidad Panamericana, validated by the Social Affairs Committee and approved by the Governing Council through resolution CR 98-22 on 15 November 2022.

**Informed Consent Statement:** Informed consent was obtained from all subjects involved in the study.

**Data Availability Statement:** Data is contained within the article.

**Acknowledgments:** This work was supported in part via a collaboration with the REDTPI4.0 CYTED program. This work is part of the project "Tecnologías de la Industria 4.0 en Educación, Salud, Empresa e Industria" developed by Universidad Indoamérica.

**Conflicts of Interest:** The authors declare no conflicts of interest.

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
