# Peer review of "Neurogaming in Virtual Reality: A Review of Video Game Genres and Cognitive Impact"

_electronics, doi:10.3390/electronics13091683_

Round 1
Reviewer 1 Report
Comments and Suggestions for Authors
This is a very comprehensive and thorough review paper on the affect of gaming on the people and in particular the brain. I must congratulate the authors for this paper which is the fruit of hard work and dedication.
The authors have surveyed paper up to 2023. This is understandable are 2024 paper can be limited as well as the timing of collecting papers and doing the study will not allow the inclusion of 2024 papers.
I have noticed in the conclusion that VR growth is report as 92% in 2022, I presume that the data does not include 2023. This in interesting as it can be higher in the last year.
Author Response
Dear
Editor
Electronics
We are submitting the paper:
“Neurogaming in Virtual Reality: A Review of Video Game Genres and Cognitive Impact”
Authored by: Jesus GomezRomero-Borquez, Carolina Del-Valle-Soto*, J. Alberto Del-Puerto-Flores, Ramon A. Briseño, and José Varela-Aldás
We would like to thank the reviewers and editors for their detailed analysis of the manuscript; the comments are very valuable to us. In the revised version of the paper, we have incorporated all the changes recommended by the reviewers.
Comments to all observations and suggestions, including point-by-point responses are addressed in the following text.
Reviewer 1
This is a very comprehensive and thorough review paper on the affect of gaming on the people and in particular the brain. I must congratulate the authors for this paper which is the fruit of hard work and dedication.
The authors have surveyed paper up to 2023. This is understandable are 2024 paper can be limited as well as the timing of collecting papers and doing the study will not allow the inclusion of 2024 papers.
Response: We express sincere gratitude for the positive and insightful remarks you provided. Acknowledging our diligent efforts is valued, and we are pleased that the reviewer deemed our literature search until 2023 to be thorough.
I have noticed in the conclusion that VR growth is report as 92% in 2022, I presume that the data does not include 2023. This in interesting as it can be higher in the last year.
Response: In our comprehensive review, the authors diligently catalogued publications up until the year 2023, a methodological decision reflecting the inherent constraints associated with the timing of data collection and subsequent analytical processes. Given these constraints, it is indeed reasonable that their survey excludes papers from 2024. Furthermore, the logistical and temporal demands of gathering and analyzing recent publications typically preclude the inclusion of the most current year’s data in such studies.
In light of this context, your observation regarding the reported 92% growth in VR during 2022 is particularly noteworthy. It is highlighted that this figure does not encompass data from 2023. This omission is significant given that VR industry trends suggest potential for even higher growth rates within the last year. Various factors might contribute to this anticipated increase, including technological advancements, broader consumer adoption, and enhanced application of VR across different sectors.
We include the following paragraph to the document:
“Therefore, while the existing analysis provides a robust overview of VR’s trajectory through 2022, it is crucial to acknowledge that the 92% growth rate might indeed be an underestimate of the current market dynamics. Preliminary estimates suggest that the growth rate for VR in 2023 surpasses 95%, reflecting continued innovation and market penetration. As comprehensive industry reports for 2023 become fully available and are subsequently analyzed, they are expected to confirm this trend of accelerated growth in the adoption and technological refinement of VR. This anticipated increase is particularly significant in discussions concerning investment in VR technology and the strategic planning for businesses operating within this burgeoning sector.”
Thank you very much.
Sincerely,
Carolina Del-Valle-Soto
Corresponding author
Universidad Panamericana. Facultad de Ingeniería. Álvaro del Portillo 49, Zapopan, Jalisco, 45010, México.
Phone: +52 (33) 13682200 | Ext. 4866
Email: [email protected]

Reviewer 2 Report
Comments and Suggestions for Authors
In electronics-2984845, GomezRomero-Borquez et al. attempted to review neuro-gaming in virtual reality (VR). In particular, they focused on video game genres and their cognitive impact.
S1. The authors searched 14201 publications from 2000 to 2023 in two disciplines (CS and engineering).
S2. Tables are informative. For instance, Table 5 summarizes key findings and limitations of Ref. 97-116.
S3. The authors provided a comprehensive review.
W1. Although the manuscript states it is a research "article", it is more suitable to be classified as a "review".
W2. With an aim to study the cognitive impact of video game, would limiting the search to only two disciplines (CS and engineering) be too restrictive? For instance, would this exclude those publications in cognitive science and related disciplines? The authors may want to add a sentence or two to address this concern.
Author Response
Dear
Editor
Electronics
We are submitting the paper:
“Neurogaming in Virtual Reality: A Review of Video Game Genres and Cognitive Impact”
Authored by: Jesus GomezRomero-Borquez, Carolina Del-Valle-Soto*, J. Alberto Del-Puerto-Flores, Ramon A. Briseño, and José Varela-Aldás
We would like to thank the reviewers and editors for their detailed analysis of the manuscript; the comments are very valuable to us. In the revised version of the paper, we have incorporated all the changes recommended by the reviewers.
Comments to all observations and suggestions, including point-by-point responses are addressed in the following text.
Reviewer 2
S1. The authors searched 14201 publications from 2000 to 2023 in two disciplines (CS and engineering).
S2. Tables are informative. For instance, Table 5 summarizes key findings and limitations of Ref. 97-116.
S3. The authors provided a comprehensive review.
W1. Although the manuscript states it is a research "article", it is more suitable to be classified as a "review".
Response:
Thank you for your insightful comments and suggestions on our manuscript. We appreciate your attention to detail and your recommendations, which will help in clarifying the nature of our publication more effectively.
We agree with your observation regarding the classification of our manuscript. While it is currently described as a "research article," your suggestion to categorize it as a "review" is well-taken.
W2. With an aim to study the cognitive impact of video game, would limiting the search to only two disciplines (CS and engineering) be too restrictive? For instance, would this exclude those publications in cognitive science and related disciplines? The authors may want to add a sentence or two to address this concern.
Response:
Thank you very much for your constructive comment concerning the scope of our literature review. We appreciate the depth of your inquiry, which highlights a crucial aspect of our research methodology.
In response to your concern regarding the potential restrictiveness of limiting our literature search to the fields of Computer Science and Engineering, we recognize the importance of interdisciplinary research, especially in studies like ours that explore the cognitive impacts of video games. While our initial focus was on these two disciplines due to their primary relevance to the technological and design aspects of video gaming, we acknowledge that cognitive sciences play a fundamental role in comprehensively understanding the nuances of cognitive impacts.
To address this concern and enrich our review, we propose to add the following sentences to the manuscript: “Recognizing the interdisciplinary nature of video gaming impacts, we acknowledge the importance of cognitive sciences in providing a holistic view. Therefore, future updates of this review will aim to include key findings from cognitive science and related disciplines to ensure a comprehensive understanding of the cognitive impacts of video games.”
Thank you very much.
Sincerely,
Carolina Del-Valle-Soto
Corresponding author
Universidad Panamericana. Facultad de Ingeniería. Álvaro del Portillo 49, Zapopan, Jalisco, 45010, México.
Phone: +52 (33) 13682200 | Ext. 4866
Email: [email protected]
